# In praise of stubbornness:
# The Case for Cognitive-Dissonance Aware Continual Update of Knowledge in LLMs

## Abstract

Despite remarkable capabilities, large language models (LLMs) struggle to continually update their knowledge without catastrophic forgetting. In contrast, humans effortlessly integrate new information, detect conflicts with existing beliefs, and selectively update their mental models. This paper introduces a cognitive-inspired investigation paradigm to study knowledge updating in LLMs. We implement two key components inspired by human cognition: (1) *Dissonance and Familiarity Awareness*, analyzing model behavior to classify information as novel, familiar, or dissonant; and (2) *Targeted Network Updates*, which track neural activity to identify frequently used (*stubborn*) and rarely used (*plastic*) neurons. Through carefully designed experiments in controlled settings, we uncover a number of empirical findings demonstrating the potential of this approach. First, dissonance detection is feasible using simple activation and gradient features, suggesting potential for cognitive-inspired training. Second, we find that non-dissonant updates largely preserve prior knowledge regardless of targeting strategy, revealing inherent robustness in LLM knowledge integration. Most critically, we discover that dissonant updates prove catastrophically destructive to the model's knowledge base, indiscriminately affecting even information unrelated to the current updates. This suggests fundamental limitations in how neural networks handle contradictions and motivates the need for new approaches to knowledge updating that better mirror human cognitive mechanisms.

## 1 Introduction

Humans effortlessly update their knowledge as they experience the world. They seamlessly integrate new information, ignore redundant stimuli, and actively resolve conflicts with existing beliefs before updating their mental models. This cognitive flexibility stems from several key abilities. Humans exhibit (1) *selective attention*, focusing on novel or relevant information while filtering out irrelevant or familiar stimuli (Posner et al., 1990; Petersen & Posner, 2012; Desimone et al., 1995; Ranganath & Rainer, 2003). They readily (2) *detect conflicts* (Croyle & Cooper, 1983) between new information and existing knowledge and actively engage in resolving them, a process known in psychology as cognitive-dissonance (Festinger, 1957; Van Veen et al., 2009). Moreover, their brains exhibit a form of (3) *adaptive plasticity*, allowing for updates to neural networks that can incorporate new information while often preserving existing knowledge. While the exact mechanisms are still being investigated, this process seems to balance the stability of well-established knowledge with flexibility in the face of new or uncertain information (McClelland et al., 1995; Behrens et al., 2007).

Despite demonstrating remarkable capabilities across various tasks, Large Language Models (LLMs) are still far from such learning abilities. Current LLMs face significant challenges in real-world deployment and long-term utility due to their static nature and training paradigms. They suffer from catastrophic forgetting (Kirkpatrick et al., 2017a; Kemker et al., 2018; Li et al., 2022; Luo et al., 2024; Kotha et al., 2024), where incorporating new information often leads to the erasure of previously learned knowledge. Furthermore, LLMs engage during training in indiscriminate learning, passively accepting all training data, even when it contradicts what they already learned. Despite emergent sparsity (Jaiswal et al., 2023; Mirzadeh et al., 2024), knowledge in LLMs follows backpropagation and the objective function, with no explicit mechanism for targeted knowledge

Table 1: Taxonomy of Incremental Learning Approaches. See Appendix.A for an extended version

| Examples | Incremental Type | Memory Usage | Task Awareness | Weight Plasticity | Architecture | Conflict Detection | Update Mechanism |
|---|---|---|---|---|---|---|---|
| iCaRL (Rebuffi et al., 2017) | Class-incremental | Replay | Task-Agnostic | Fixed | Fixed | No | Rehearsal |
| EWC (Kirkpatrick et al., 2017b) | Task-incremental | None | Task-Aware | Selective | Fixed | No | Regularization |
| Progressive Nets (Rusu et al., 2016) | Task-incremental | None | Task-Aware | Fixed | Expanding | No | New Subnetworks |
| DEN (Yoon et al., 2017) | Task-incremental | None | Task-Aware | Selective | Expanding | No | Selective Expansion |
| GEM (Lopez-Paz & Ranzato, 2017) | Task-incremental | Replay | Task-Aware | Constrained | Fixed | No | Constrained Optimization |
| ROME (De Cao et al., 2021) | Fact-incremental | None | Fact-Aware | Localized | Fixed | No | Rank-One Update |
| OWM (Zeng et al., 2019) | Task-incremental | None | Task-Aware | Orthogonal | Fixed | No | Orthogonal Projection |
| PackNet (Mallya & Lazebnik, 2018) | Task-incremental | None | Task-Aware | Selective | Fixed | No | Weight Masking |
| HAT (Serra et al., 2018) | Task-incremental | None | Task-Aware | Selective | Fixed | No | Attention Masking |
| **This paper** | Fact-incremental | None | Conflict-Aware | Selective | Fixed | Yes | Neuron-Specific Update |

storage or retrieval. This results in a situation where all weights are potential candidates for storing knowledge, necessitating comprehensive retraining to properly incorporate new information.

In this work, we embark on a systematic empirical investigation of how LLMs handle knowledge updates, drawing inspiration from human cognitive traits. Through carefully controlled experiments, we examine (1) the feasibility of *Dissonance Awareness*, i.e. whether it is possible to correctly classify facts into novel, familiar, and dissonant using features extracted from the LLM. We also investigate the benefits of (2) *Adaptive Plasticity* by studying how different neuron targeting strategies affect knowledge retention and update. For this, we develop a simple method for tracking historical neuron usage to identify "plastic" (rarely used) and "stubborn" (previously used) neurons, allowing us to study how knowledge updates affect different regions of the model's parameter space. This experimental framework lets us systematically investigate fundamental properties of knowledge integration in LLMs.

Our investigation reveals a fundamental distinction in how LLMs handle knowledge updates: the case of non-dissonant updates (adding entirely new knowledge) versus dissonant updates (modifying existing associations). While prior work has focused mostly on editing individual factual associations within LLMs (Meng et al., 2022a; Mitchell et al., 2022; Meng et al., 2022c) or preserving knowledge across distinct tasks as in continual learning (Rebuffi et al., 2017; Kirkpatrick et al., 2017b; Mallya & Lazebnik, 2018), our controlled experiments take a different approach. We systematically study how the placement of new knowledge in the network's parameter space affects both the integration of that knowledge and its impact on existing, unrelated knowledge. As shown in 1, this positions our work uniquely: rather than proposing new editing or continual learning methods, we reveal fundamental properties about how LLMs handle knowledge integration in both dissonant and non-dissonant scenarios. Critically, our experimental design allows us to precisely track the impact of updates on a controlled set of initial knowledge, providing clear visibility into how different update strategies affect unrelated information.

**Key takeaways.** This leads us to uncover several fundamental properties of LLM knowledge updating: (i) *dissonance awareness* is feasible using simple model features, suggesting potential for cognitive-inspired training; (ii) LLMs show inherent robustness when incorporating non-dissonant information, largely preserving prior knowledge regardless of targeting strategy; (iii) avoiding heavily-used (stubborn) neurons during updates further improves this robustness, motivating *adaptive plasticity* in this scenario; (iv) regions of the network heavily used during pre-training are particularly effective at incorporating new knowledge, extending lottery ticket hypothesis findings (Frankle & Carbin, 2019) to language models; and most critically, (v) dissonant updates prove catastrophically destructive to unrelated knowledge, suggesting fundamental limitations in how neural networks handle contradictions: while some of our targeted update strategies show comparable performance to existing editing methods like ROME and MEMIT, all approaches fundamentally struggle with dissonant updates, suggesting the need for fundamentally different mechanisms.

**Implications.** These findings point to concrete opportunities such as the feasibility of dissonance awareness, and the benefits of adaptive plasticity in case of non-dissonant updates. But they also reveal fundamental challenges when handling contradictory information. Current approaches essentially attempt to erase and replace old knowledge - a process we show leads to catastrophic forgetting of even unrelated information. But this contrasts sharply with human cognition, where we maintain both old and new knowledge with appropriate temporal context. Consider how humans handled learning that Pluto was no longer classified as a planet: rather than erasing our previous understand-

ing, we maintained both pieces of knowledge, understanding their historical context and why the classification changed. Our experiments motivate the exploration of future fundamentally different mechanisms for handling contradictions - ones that can maintain and contextualize conflicting information rather than attempting to overwrite it.[1]

## 2 Dissonance-aware Targeted Knowledge Update

The core of our approach involves (1) awareness concerning the type of information the model ingests, which is then used to (2) selectively target sparse portions of the LLMs for incremental updates. Both rely on the extraction of activations and gradients.

### 2.1 Extraction of historical activations and gradients

We maintain an aggregate profile of neuronal activity by accumulating activations and gradients for each neuron at every training step. Specifically, for each neuron $n$ in the Transformer blocks—including feed-forward (MLP) layers and attention projections (Key, Query, Value matrices)—we compute $H\hat{G}_n$, the cumulative *historical gradient* magnitude over time, and $H\hat{A}_n$, the cumulative *historical activation* magnitude over time. To mitigate scale differences across layers, we also experiment with layer-wise normalization of activations and gradients before accumulation. Precise notation and computation methods are detailed in Appendix B.

This historical activity data enables us to classify neurons as "plastic" or "stubborn" based on their past usage, which is useful for our targeted network updates. We use the historical data also to normalize the input features when classifying facts as we see next.

### 2.2 Dissonance and Novelty Awareness

We cast our classification problem on three classes: for a given input sequence $X$, decide if it is *Novel* (and should be integrated by the Transformer), *Familiar* (and can be ignored), or *Dissonant* (thus likely requiring proper resolution).

We design a simple classifier that leverages activation and gradient information to assess the nature of new information. For any input sequence $X$, we first perform a forward pass to obtain its *current activations* and a backward pass to obtain its *current gradients* (without updating the model weights). Since the goal is to assess feasibility using easy-to-compute features and lightweight methods that could be integrated into large-scale models, we extract for each layer the mean, standard deviation, minimum, maximum, and quartiles (Q1, Q2, Q3) of the activations and gradients, eventually first normalized by historical activations and gradients. We perform ablation studies to assess the importance of different features and employ feature importance analyses to understand which aspects contribute most to the classifier's performance. We evaluate our ability to classify facts in Sec. 3.2. Despite using simple classifiers like Random Forests and SVMs, we achieve high accuracy, opening the way for future integration of dissonance awareness into LLM training pipelines.

### 2.3 Targeted Neuron Updates

Building upon the historical tracking of neural activity, we implement targeted network updates to incorporate new knowledge into the model's parameters while preserving existing information. We design four main types of targeted updates, which we experimentally evaluate. During training on new information, we perform standard forward and backward passes to compute the loss and gradients. Before the optimizer step, we modify the gradients to freeze certain neurons. Specifically, given the gradients for all parameters of a given layer, we zero-out those that do not belong to the selected set of neuron and corresponding weights, defined as plastic, stubborn, candidate and specific, as described below. This process effectively freezes the weights of non-selected neurons, allowing for targeted updates to specific parts of the model. By varying the choice of selected neurons, we control how new information is integrated into the model while managing its impact on existing knowledge. Next, we introduce strategies to select which neurons and weights to update:

---

[1]Anonymized code available at `https://figshare.com/s/81f7108d823b5e08e8ec`

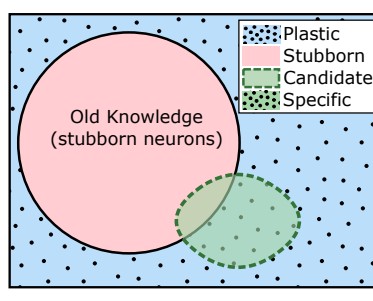
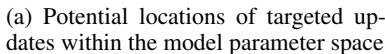

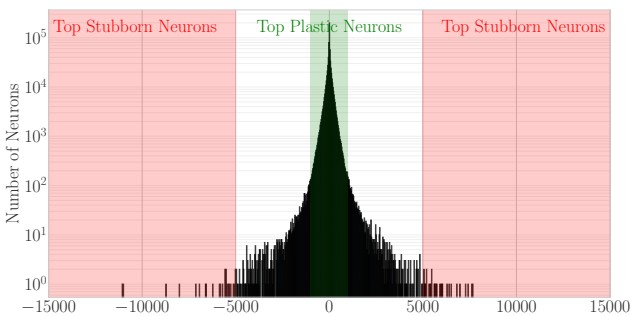

(a) Potential locations of targeted updates within the model parameter space

(b) Distribution of cumulated gradients (during gpt2-xl training)

Figure 1: *Targeted neuron updates.* The historical activity of neurons during previous training is used to identify localized areas where to store future knowledge, according to four strategies.

Figure 1a illustrates the conceptual relationship between the various neuron updates strategies within the model's parameter space.

**Plastic Neurons.** Neurons underutilized during past model updates. To identify them, we rank neurons by increasing historical gradient values and select the top $N$ neurons with the lowest cumulative gradients:

$$\mathcal{N}_{\text{plastic}} = \{n \mid \text{rank}(H\hat{G}_n) \leq N\},$$

where $H\hat{G}_n$ is the historical gradient for neuron $n$, accumulated over all prior training. By targeting underutilized neurons, we aim to integrate new knowledge while minimizing interference with existing information.

**Stubborn Neurons.** Neurons that accumulated high historical gradients, indicating significant involvement in previous learning. We rank neurons by decreasing historical gradient values and select the top $N$ neurons:

$$\mathcal{N}_{\text{stubborn}} = \{n \mid \text{rank}(H\hat{G}_n) > |\mathcal{N}| - N\},$$

where $|\mathcal{N}|$ is the total number of neurons, and $H\hat{G}_n$ is the historical gradient for neuron $n$. Updating stubborn neurons allows us to test the model's capacity for knowledge integration and assess the potential risks of overwriting existing information.

**Candidate Neurons.** These neurons are relevant for encoding new information: to identify them, we perform a single back-propagation pass on the new input data, without updating the model weights. We then rank neurons based on the magnitude of these gradients and select the top $N$:

$$\mathcal{N}_{\text{candidate}} = \{n \mid \text{rank}(G_n^{\text{new}}) > |\mathcal{N}| - N\},$$

where $G_n^{\text{new}}$ is the gradient for neuron $n$ obtained from the back-propagation pass on the new input data. Targeting candidate neurons focuses updates on areas of the network that are most relevant to the new information, as suggested by the back-propagation process.

**Specific Neurons.** To identify neurons capable of storing new information while avoiding interference with existing knowledge, we first: (1) identify stubborn neurons $\mathcal{N}_{\text{stubborn}}$, using $N$ as defined earlier; we next (2) rank all neurons based on the magnitude of their gradients $G_n^{\text{new}}$ obtained from a single back-propagation pass on the new data, without updating model weights; finally, (3) we select few specific neurons, by choosing the top $N$ neurons that are not in $\mathcal{N}_{\text{stubborn}}$:

$$\mathcal{N}_{\text{specific}} = \text{Top}_N(\mathcal{N}_{\text{all}} \setminus \mathcal{N}_{\text{stubborn}}),$$

where $\mathcal{N}_{\text{all}}$ is the set of all neurons ranked by their gradient magnitudes. This last approach ensures that we select neurons that are most relevant to the new information (high gradient) while explicitly avoiding those that are crucial for existing knowledge (stubborn neurons).

## 3 EXPERIMENTAL EVALUATION

We discuss our (1) experimental setup, to evaluate (2) cognitive dissonance-awareness, as well as (3) continual knowledge update.

### 3.1 EXPERIMENTAL SETUP

**Dataset** We use the COUNTERFACT dataset (Meng et al., 2022b) as our primary data source, containing both facts and counterfacts.[2]. This dataset, with approximately 17,000 facts, allows us to test models' handling of conflicting knowledge and addition of potentially known information[3], two key aspects of our dissonance-aware approach. To address the lack of truly novel facts in COUNTERFACT, we generate additional data using GPT-3.5. We transform existing statements into plausible yet fictitious information, maintaining structural similarity while introducing novel content. For example, "Danielle Darrieux's mother tongue is French" becomes "Sylvan Myrthil's mother tongue is Sylvan" (see Appendix C.1 for details).

For our dissonance awareness experiments, we construct a balanced dataset comprising 1,000 samples each of familiar, conflicting, and novel facts. When using the pre-trained GPT-2-small model, we adjust the familiar class to 600 samples due to the limited number of known facts extracted from the model's pre-training. For our targeted update experiments, we use 5-fold cross-validation varying each time the sets of old and new facts but keeping the following proportions: 2000 old facts vs. 1000 new facts. For conflicting updates, we also test with 10 and 100 new facts.

**Models** We employ both GPT-2-small and GPT-2-xl for their accessibility, and to facilitate reproducibility and scale impact analysis. However, the dataset size ($\simeq$17,000 facts) limits full stress-testing of larger models like GPT-2-xl (less visible catastrophic forgetting compared to compressed models). As a result, the effects of our experiments are most clearly observed with GPT-2-small, on which we focus most in the main body of this paper, deferring GPT-2-xl results to the Appendix.

We implement experiments using Hugging Face Transformers on NVIDIA GPUs. Before setting learning rates and epochs, we conduct a search for optimal hyperparameters that allow effective learning of facts (see App. D.2 for an example for GPT-2-xl). We perform the search based on the ability to correctly learn 10,000 facts from the dataset. More detailed results and our implementation are available in the code repository.

### 3.2 DISSONANCE AWARENESS

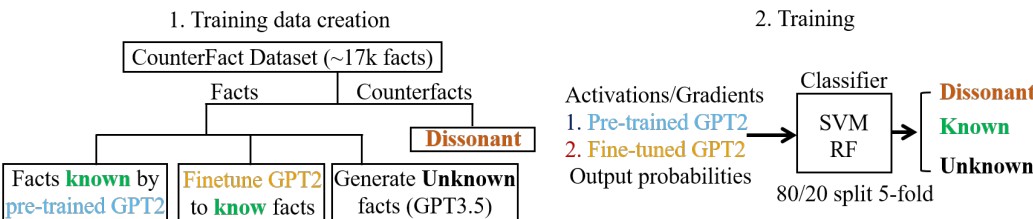

Figure 2: Classifier pipeline from data creation to classification.

**Settings.** Our first goal is to evaluate the ability to discriminate *familiar*, *novel*, and *conflicting* information using the readily-available[4] simple features we extract from the models during the forward and backward passes. As schematized in Fig. 2, we do so by relying on simple classifiers (random forests and SVMs), contrasting two scenarios for the *input features*: (1) a GPT-2 model fine-tuned on 1000 facts (the knowns), and (2) a GPT-2 pre-trained model (using its 600 extracted known facts as known class samples).

For each scenario, we compile a balanced dataset with equal examples per class (familiar, novel, conflicting). To create novel facts, we employ GPT-4 with carefully designed prompts, using the structure of known facts (subject, relation, object) as templates, replacing key elements with fictitious (but plausible) information. This method ensured structural similarity to known facts while

---

[2]While this dataset allows us to test models' handling of conflicting knowledge, we acknowledge its limitations in representing more complex real-world knowledge, a limitation which we plan to address in the future

[3]For instance, general facts that pre-trained models likely were exposed to during training.

[4]We explore the use of model output-only features in Appendix. C.5 showing that using output probabilities as feature is also successful.

maintaining novelty. Appendix C.1 provides detailed prompts and examples (full datasets will be made available upon acceptance).

**Classification performance**  We extracted activations (A) and gradients (G) as described in Appendix B, and experimenting with A, G and A+G as input feature sets, using raw (R), per-layer (L) and historical (H) normalization strategies. As classifiers, we employ Random Forest (RF) and Support Vector Machines (SVM), optimizing hyperparameters using Bayesian search with 5-fold cross-validation. For clarity, we report the best results for each combination in Table 2 (average and standard deviation accuracy over the 5-folds) and defer the full results and ablation study to Table 5 in the appendix for the interested reader.

Models consistently achieve high performance. Using features from the finetuned model reaches as high as (99.5%), but also using features from a pre-trained model still achieves decent performance (94.7%). Interestingly, combining activations and gradients consistently outperformed using either feature set alone, with a slight advantage of SVM over RF. Also, historical normalization helps SVM, but does not provide benefits for RF.

Table 2: Classification Results

| Scenario | Classifier | Accuracy |
|---|---|---|
| Fine-tuned | SVM (A+G, H) | 0.995 (0.001) |
|  | RF  (A+G, R) | 0.988 (0.001) |
| Pre-trained | SVM (A+G, H) | 0.947 (0.004) |
|  | RF  (A+G, R) | 0.928 (0.012) |

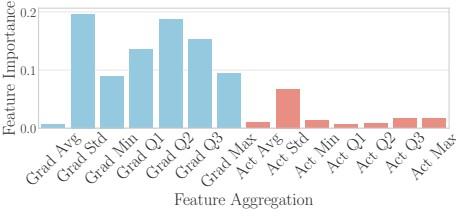

(a) Finetuned model

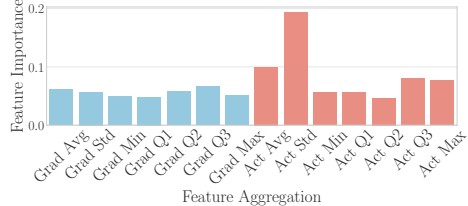

(b) Pretrained model

Figure 3: *Dissonance awareness*. Feature importance grouped by feature type (notice the higher importance of gradient-related features for finetuned models).

**Feature importance**  While a full analysis of feature explainability is outside the scope of this paper, we further seek to observe structural differences in the learning process by comparing feature importance, using the feature importance scores derived from the random forest algorithm.

To further shed light on such difference, Fig. 3 opposes feature importance in both cases, focusing on Activation versus Gradient-related features. It turns out that in the finetuned scenario, gradient-based features are substantially more important. This is likely due to the fact that finetuning the models on these facts has somewhat overfit them leading to gradients that are more discriminative: e.g. a clearly null gradient for known facts and a clearly high one for unknown ones. For the pretrained scenario, however, which is the most likely case in a real case scenario, both activation and gradient features contribute significantly, suggesting that for long-term knowledge, both internal representations and learning dynamics should be mixed in order to achieve good classification. Appendix.C.3 expands this analysis by focusing on transformer block importance instead.

Finally, deferred to the appendix, comparing the performance of different normalization strategies for the pretrained model scenario using both activations and gradients (Table 5), we found that although normalization slightly helps, historical normalization does not seem to be crucial, since it was only slightly helpful for Random Forest classifiers.

**Key findings** Overall, despite the simplicity of our features, the results demonstrate the feasibility of distinguishing between familiar, novel, and conflicting information, even in the challenging case of using pre-trained models, providing the needed foundation for dissonance-aware updates, which we explore in the next experiments.

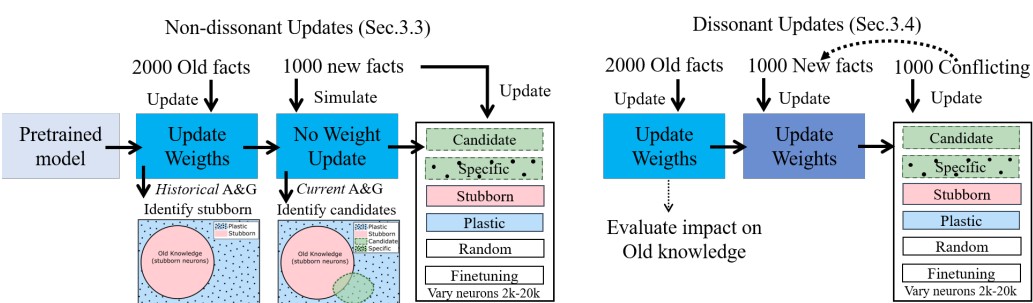

Figure 4: Overview of our controlled experiments in case of dissonant and non-dissonant updates.

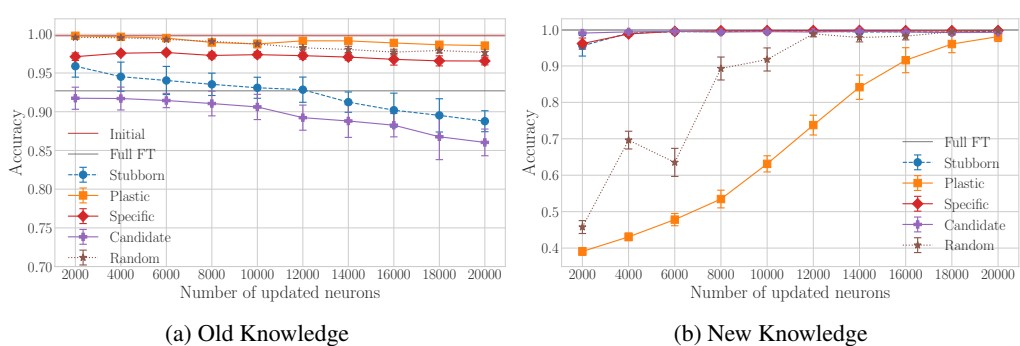

(a) Old Knowledge

(b) New Knowledge

Figure 5: *Non-dissonant updates*: Old vs new knowledge for targeted updates on GPT-2-small.

## 3.3 NON-DISSONANT UPDATES

**Settings.** We now investigate how LLMs handle *non-dissonant updates* using our different strategies as experimental tools. In our experiments, schematized in Fig. 4 (left), we evaluate the incorporation of non-conflicting facts into GPT-2-small and GPT-2-xl. The pipeline consists of (i) training on 2,000 initial facts (old) while collecting historical gradients to identify stubborn neurons, (ii) simulating updates with 1,000 new facts (new) to collect current gradients for candidate identification and (iii) applying different neuron selection strategies to update the model with these 1,000 facts. Note that while we track 2,000 facts as proxy for old knowledge, this represents a smaller fraction of GPT-2-xl's total knowledge compared to GPT-2-small, limiting our visibility into effects on other untracked pre-trained knowledge.

For one particular experiment inspired by the lottery ticket hypothesis (Frankle & Carbin, 2018), we used 10,000 separate facts for gradient extraction before training a fresh model on the 2,000+1,000 facts setup described above (details in Appendix D.1). Due to space constraints, we defer comprehensive ablation studies and additional experiments to Appendix D.

**Results.** Fig. 5 presents the accuracy of various neuron update strategies on old and new knowledge for GPT-2-small, including error bars representing standard deviations over five runs. We observe that simple fine-tuning leads to a degradation in performance on old knowledge, dropping to approximately 93% accuracy. In contrast, updating plastic neurons helps preserve old knowledge, with accuracy remaining above 98% even when using up to 20,000 neurons. Random neuron selection exhibits a similar behavior. However, using candidate or stubborn neurons results in slightly more degradation to old knowledge. Interestingly, the specific neurons strategy strikes a balance between learning new knowledge efficiently and preserving old knowledge. It achieves higher accuracy on new knowledge with fewer neurons while minimizing the impact on old knowledge.

To visualize the trade-offs between learning new knowledge and preserving old knowledge, Fig. 6 shows scatter plots of old knowledge accuracy versus new knowledge accuracy for various neuron thresholds ranging from 2,000 to 20,000 neurons. The plots illustrate that targeting plastic neurons tends to preserve old knowledge but may require more neurons to achieve high accuracy on new

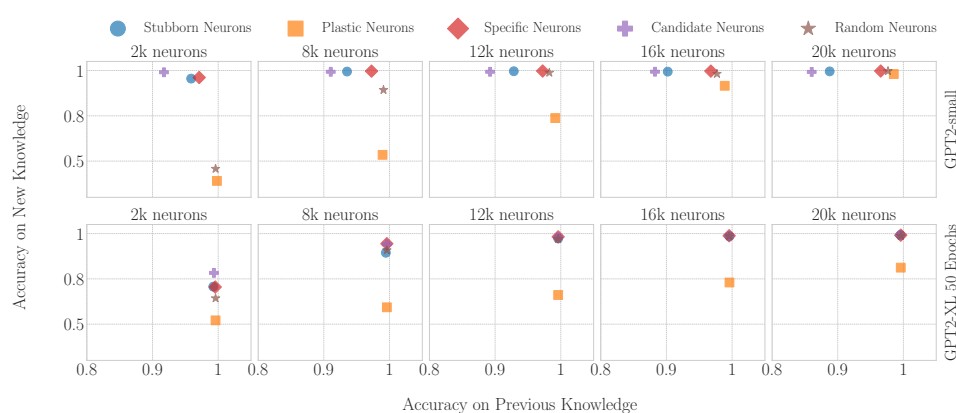

Figure 6: *Non-dissonant updates*: Scatter plot of old (x) vs new (y) knowledge for different strategies and number of neurons. GPT-2-small (top row) and GPT-2-xl (bottom) base configuration.

knowledge. In contrast, targeting specific neurons allows for efficient learning of new knowledge with fewer neurons while maintaining acceptable levels of old knowledge retention.

We conducted similar experiments on GPT-2-xl, where our 2,000 tracked facts represent a much smaller portion of the model's knowledge. With this larger capacity, interference with tracked facts becomes naturally less likely, explaining why all strategies show good preservation of our monitored knowledge. As shown in the bottom row of Fig. 6, all strategies generally preserve old knowledge in GPT-2-xl ; however, they differ in their ability to integrate new knowledge efficiently. Detailed analyses, including the effects of varying learning rates and neuron counts, are provided in Appendix D.3. Overall, we found that observing similar effects in GPT-2-xl required adjusting either the learning rate, the number of neurons allocated for updates, or learning longer. The latter (50 epochs as opposed to 5) is the option we've used in Fig. 6.

An intriguing result is that targeting stubborn, candidate, or specific neurons allows the model to learn new knowledge using fewer parameters compared to targeting plastic neurons. This finding resonates with the existence of winning subnetworks, as suggested by the lottery ticket hypothesis (Frankle & Carbin, 2018). It implies that certain subnetworks within the model are more conducive to integrating new information, compared to others. We conduct further experiments confirming this lottery ticket hypothesis in App D.1.

**Key Findings.**    These experiments reveal that LLMs show remarkable robustness when incorporating non-dissonant information, as long as heavily-used (stubborn) neurons are avoided. Updating plastic neurons helps preserving old knowledge but requires more parameters (or time) to achieve high accuracy on new knowledge. Targeting specific neurons offers a balanced approach, enabling efficient knowledge integration with minimal impact on existing information.

## 3.4    DISSONANT UPDATES

**Settings.**    We now examine how LLMs handle conflicting (dissonant) information. As shown in Fig. 4 (right), after training on 2,000 old facts and 1,000 new facts, we introduce 1,000 conflicting updates that contradict the previously learned new facts. The targeted neuron update strategies defined earlier are applied to assess their effectiveness in handling conflicting information, measuring impact on both the conflicting facts and the unrelated old knowledge.

**Results.**    We illustrate the performance of various strategies when editing GPT-2-small with 1,000 facts in Fig. 7. Here, old knowledge refers to the original facts, new knowledge corresponds to the conflicting (counterfactual) facts, and generalization measures the model's accuracy on paraphrased versions of the new facts. Surprisingly, we find that dissonant updates are highly destructive to the retention of old knowledge, regardless of the neuron update strategy employed. Even when updating

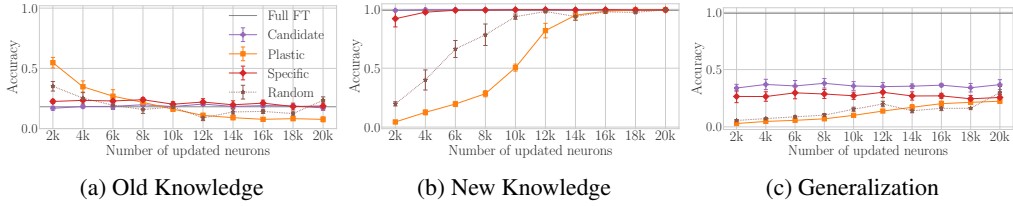

(a) Old Knowledge      (b) New Knowledge      (c) Generalization

Figure 7: *Dissonant updates*: Impact of 1000 dissonant facts and GPT-2-small

plastic neurons, which are presumed to be underutilized and thus less likely to interfere with existing knowledge, we observe significant degradation in the model's ability to recall old facts.

Given the observed difficulty of simultaneously editing 1,000 facts, we conducted additional experiments where we edited 100 and 10 facts. The results, detailed in Appendix E.1, indicate that while the impact on old knowledge retention is less severe when editing fewer facts, the destructive effect remains prominent. Notably, the performance of state-of-the-art model editing methods such as ROME (Meng et al., 2022a) and MEMIT (Meng et al., 2022c) also deteriorates when applied to multiple sequential edits, as opposed to the single-edit evaluations typically reported in the literature. While our primary focus is not on developing new model editing techniques, we leverage EasyEdit (Wang et al., 2023) to benchmark the above existing methods under our multi-fact experimental conditions.

Table 3 summarizes the performance of different strategies and editing methods. Some of our targeted update strategies obtain a higher harmonic mean compared to ROME and MEMIT, but the approaches are not directly comparable since they explore different regions of the pareto front, balancing new knowledge acquisition and old knowledge retention, as self-explained with colors and rankings in the table.

Finally, we also performed experiments with GPT-2-xl under various conditions, deferred to Appendix E.3 for space constraints. Overall, similarly to the non-dissonant case, GPT-2-xl fails to learn the new conflicting knowledge effectively. Surprisingly though, despite not learning new knowledge, and despite having much more parameters, *GPT-2-xl also experiences significant degradation in old knowledge retention* – further confirming the catastrophic nature of dissonant updates, even for such a larger model (See Fig. 14).

**Key Findings.** Dissonant updates pose a significant challenge, as they are destructive to prior unrelated knowledge, regardless of model size and even when targeting unused neurons. This underscores the importance of dissonance awareness to detect and appropriately handle conflicting information during continual learning. Our results motivate the integration of dissonance classifiers directly into the update or training of large language models. Thus, developing dedicated conflict resolution methods remains an essential direction for future work.

## 4 DISCUSSION AND CONCLUSIONS

### 4.1 LESSONS LEARNED

**Fundamental Properties of Knowledge Updates:** Our results reveal striking differences between dissonant and non-dissonant updates. Non-dissonant updates show remarkable robustness, naturally preserving existing knowledge regardless of strategy (as long as stubborn neurons are avoided). In contrast, dissonant updates prove catastrophically destructive - with all tested strategies, accuracy on unrelated knowledge dropped below 60% when updating just 10 to 100 conflicting facts.

**Feasibility of Dissonance Detection:** LLMs encode clear signatures that distinguish between novel, familiar, and dissonant information. Simple classifiers using either activation and gradient features (or output probabilities) achieve more than 95% accuracy with pre-trained models and 99% with finetuned models, suggesting potential for cognitive-inspired training pipelines that could clean the data from conflicting information before feeding them for training.

Table 3: Comparison of targeted neuron update strategies vs knowledge-editing literature, with a gradient from 0 (red) to 1 (green). Top-1,2 strategies annotated for all metrics and sample sizes.

| Samples | Strategy | Old (Unrelated) | New (Reliability) | Generalization | Harmonic Mean |
|---|---|---|---|---|---|
| 10 | Full Finetune | 0.107 (0.082) | 1.000 (0.000) [1] | 0.576 (0.117) | 0.222 (0.116) |
| | MEMIT(Meng et al., 2022c) | 0.962 (0.079) [1] | 0.000 (0.000) | 0.000 (0.000) | 0.000 (0.000) |
| | ROME(Meng et al., 2022a) | 0.891 (0.085) | 0.240 (0.182) | 0.180 (0.179) | 0.236 (0.235) |
| | 8k Candidate | 0.596 (0.106) | 0.988 (0.024) [2] | 0.644 (0.128) [2] | 0.690 (0.058) [1] |
| | 20k Candidate | 0.430 (0.134) | 1.000 (0.000) [1] | 0.656 (0.125) [1] | 0.597 (0.116) |
| | 8k Specific | 0.638 (0.138) | 0.964 (0.039) | 0.512 (0.238) | 0.600 (0.183) |
| | 8k Stubborn | 0.622 (0.110) | 0.972 (0.030) | 0.544 (0.169) | 0.643 (0.103) [2] |
| | 8k Plastic | 0.909 (0.039) [2] | 0.020 (0.040) | 0.000 (0.000) | 0.000 (0.000) |
| | 8k Random | 0.827 (0.083) | 0.380 (0.132) | 0.092 (0.094) | 0.277 (0.098) |
| 100 | Full Finetune | 0.238 (0.019) | 0.998 (0.003) [2] | 0.434 (0.089) | 0.398 (0.041) |
| | MEMIT(Meng et al., 2022c) | 0.976 (0.008) [1] | 0.004 (0.005) | 0.010 (0.007) | 0.003 (0.007) |
| | ROME(Meng et al., 2022a) | 0.431 (0.108) | 0.300 (0.054) | 0.150 (0.036) | 0.240 (0.045) |
| | 8k Candidate | 0.542 (0.035) [2] | 0.969 (0.033) | 0.462 (0.081) [1] | 0.591 (0.054) [1] |
| | 20k Candidate | 0.463 (0.032) | 0.999 (0.002) [1] | 0.447 (0.083) [2] | 0.552 (0.052) [2] |
| | 8k Specific | 0.531 (0.030) | 0.760 (0.063) | 0.263 (0.027) | 0.426 (0.024) |
| | 8k Stubborn | 0.530 (0.054) | 0.936 (0.048) | 0.398 (0.064) | 0.547 (0.063) |
| | 8k Plastic | 0.433 (0.029) | 0.059 (0.014) | 0.028 (0.017) | 0.052 (0.025) |
| | 8k Random | 0.508 (0.019) | 0.193 (0.038) | 0.065 (0.025) | 0.131 (0.039) |
| 1000 | Full Finetune | 0.182 (0.007) | 0.991 (0.009) | 0.442 (0.053) [1] | 0.341 (0.016) [2] |
| | MEMIT(Meng et al., 2022c) | 0.605 (0.107) [1] | 0.198 (0.053) | 0.100 (0.016) | 0.177 (0.028) |
| | ROME(Meng et al., 2022a) | 0.152 (0.071) | 0.160 (0.093) | 0.067 (0.035) | 0.106 (0.058) |
| | 8k Candidate | 0.199 (0.014) | 0.996 (0.002) [1] | 0.380 (0.041) [2] | 0.345 (0.014) [1] |
| | 20k Candidate | 0.172 (0.018) | 0.996 (0.001) [1] | 0.369 (0.043) | 0.314 (0.028) |
| | 8k Specific | 0.240 (0.017) [2] | 0.993 (0.003) | 0.287 (0.039) | 0.345 (0.028) [1] |
| | 8k Stubborn | 0.200 (0.007) | 0.995 (0.001) [2] | 0.317 (0.024) | 0.327 (0.006) |
| | 8k Plastic | 0.218 (0.024) | 0.283 (0.026) | 0.070 (0.010) | 0.133 (0.013) |
| | 8k Random | 0.194 (0.026) | 0.663 (0.072) | 0.088 (0.008) | 0.165 (0.014) |

**Promise of differentiated plasticity:** We find that avoiding heavily-used (stubborn) neurons during *non-dissonant* updates further improves robustness, maintaining 98% accuracy on old knowledge (versus 93% with standard finetuning). Interestingly, neurons heavily utilized during pre-training prove particularly effective at integrating new knowledge, extending lottery ticket hypothesis findings (Frankle & Carbin, 2018) to language models.

## 4.2 LIMITATIONS AND FUTURE DIRECTIONS

**Experimental Control vs. Scale:** While our controlled experiments with smaller models reveal fundamental properties of knowledge updating, investigating these phenomena in larger models presents significant challenges. It is not straightforward to track the impact on their broader knowledge base.

**Dataset Limitations:** Our current findings rely on CounterFact-derived data with relatively simple factual statements. Developing larger, more diverse datasets is essential for understanding how these properties generalize to more complex forms of knowledge and conflicts.

**Neuron Classification Metrics:** Our analysis of neural plasticity relies primarily on gradient magnitudes. Future work could explore richer metrics incorporating activation patterns and network connectivity to better understand how knowledge is distributed and updated across the network.

**Beyond Binary Dissonance:** Our current investigation treats dissonance as binary, while real-world knowledge updates often involve varying degrees of conflict and different types of knowledge. Understanding how these nuances affect knowledge integration remains an open challenge.

**Towards Human-Inspired Updates:** The catastrophic nature of dissonant updates suggests we may need fundamentally different approaches to knowledge integration in LLMs. Rather than attempting to overwrite existing knowledge, future work might explore mechanisms for maintaining and contextualizing potentially conflicting information - similar to how humans maintain both historical and updated knowledge with appropriate contexts.

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

Table 4: Extended taxonomy of incremental Learning Approaches, showing some seminal work (top) and more recent literature (split into editing and continual learning).

| Examples | Incremental Type | Memory Usage | Task Awareness | Weight Plasticity | Architecture | Conflict Detection | Update Mechanism |
|---|---|---|---|---|---|---|---|
| iCaRL (Rebuffi et al., 2017) | Class-incremental | Replay | Task-Agnostic | Fixed | Fixed | No | Rehearsal |
| EWC (Kirkpatrick et al., 2017b) | Task-incremental | None | Task-Aware | Selective | Fixed | No | Regularization |
| Progressive Nets (Rusu et al., 2016) | Task-incremental | None | Task-Aware | Fixed | Expanding | No | New Subnetworks |
| DEN (Yoon et al., 2017) | Task-incremental | None | Task-Aware | Selective | Expanding | No | Selective Expansion |
| GEM (Lopez-Paz & Ranzato, 2017) | Task-incremental | Replay | Task-Aware | Constrained | Fixed | No | Constrained Optimization |
| ROME (De Cao et al., 2021) | Fact-incremental | None | Fact-Aware | Localized | Fixed | No | Rank-One Update |
| OWM (Zeng et al., 2019) | Task-incremental | None | Task-Aware | Orthogonal | Fixed | No | Orthogonal Projection |
| PackNet (Mallya & Lazebnik, 2018) | Task-incremental | None | Task-Aware | Selective | Fixed | No | Weight Masking |
| HAT (Serra et al., 2018) | Task-incremental | None | Task-Aware | Selective | Fixed | No | Attention Masking |
| MALMEN (Tan et al., 2023) | Fact-incremental | None | Fact-Aware | Localized | Fixed | No | Parameter Shift Aggregation |
| EditAnalysis (Li et al., 2023) | Fact-incremental | None | Fact-Aware | Analysis | Fixed | No | Consistency Analysis |
| D4S (Huang et al., 2024) | Fact-incremental | O(1) | Fact-Aware | Regulated | Fixed | No | Layer-Norm Control |
| Global Prototypes (Bai et al., 2024) | Task/Class-incremental | None | Task-Agnostic | Selective | Fixed | No | Global Prototype Alignment |
| NTE (Benjamin et al., 2024) | Task-incremental | None | Task-Agnostic | Selective | Fixed | No | Bayesian Ensemble |
| UPGD (Elsayed & Mahmood, 2024) | Task-incremental | None | Task-Agnostic | Selective | Fixed | No | Utility-Gated Updates |
| (Hiratani, 2024) | Task-incremental | None | Task-Aware | Selective | Fixed | No | Fisher Information |
| CLAP (Jha et al., 2024) | Class-incremental | None | Task-Aware | Selective | Fixed | No | Probabilistic Adaptation |
| VQ-Prompt (Jiao et al., 2024) | Class-incremental | None | Task-Agnostic | Fixed | Fixed | No | Discrete Prompt Selection |
| IsCiL (Lee et al., 2024) | Task-incremental | None | Task-Aware | Selective | Fixed | No | Skill-based Adaptation |
| BGS (Lee et al.) | Task/Domain/Class-incremental | Replay | Task-Aware | Selective | Fixed | Yes | Bias-Aware Update |
| SLM (Peng et al., 2024) | Task-incremental | None | Auto-detected | Selective | Fixed | No | Vector Space Retrieval |
| Train-Attention (Seo et al., 2024) | Knowledge-incremental | None | Task-Agnostic | Selective | Fixed | No | Token-Weighted Update |
| Refresh Learning (Wang et al., 2024) | Task/Class-incremental | Optional | Task-Aware | Selective | Fixed | No | Unlearn-Relearn |
| RAIL (Xu et al., 2024) | Cross-domain-incremental | None | Task-Agnostic | Selective | Fixed | No | Regression-based Update |
| SAFE (Zhao et al., 2024) | Class-incremental | None | Task-Agnostic | Selective | Fixed | No | Dual Parameter-Efficient Tuning |
| **This paper** | Fact-incremental | None | Conflict-Aware | Selective | Fixed | Yes | Neuron-Specific Update |

# APPENDIX

We now report extended material concerning the extended related work (Appendix A), the extraction of historical activations and gradients (Appendix B), as well as detailed results on dissonance awareness (Appendix C), non-dissonant updates (Appendix D) and dissonant updates (Appendix E).

# A EXTENDED RELATED WORK

In this section, we provide an extended version of Tab. 1, focusing *only* on the *most recent literature*, and showing how our work is uniquely positioned in the landscape of model editing and continual learning, the two key related branches to our work.

## A.1 CONTINUAL LEARNING

Continual Learning (CL) methods enable models to learn new tasks without catastrophically forgetting previously mastered ones (Kirkpatrick et al., 2017b). These approaches fall into three main families: memory-based methods using exemplar buffers (Rebuffi et al., 2017), knowledge distillation techniques that transfer information across model versions (Lopez-Paz & Ranzato, 2017), and regularization-based methods that constrain weight updates (Kirkpatrick et al., 2017b). To ease the understanding of this landscape, we build a taxonomy that characterizes approaches by their incremental type (task, class, or fact-based), memory requirements, update mechanisms, and architectural constraints (Table 1). This taxonomy reveals how our work is different from existing continual learning attempts: while existing methods focus on preserving knowledge across distinct tasks, none explicitly address the detection and handling of conflicting information - a key capability in human cognition that our work empirically investigates.

One of the closest old approaches is deep mind's EWC (Kirkpatrick et al., 2017b), a method designed to mitigate catastrophic forgetting in neural networks trained sequentially on distinct tasks. The core idea is to protect the most important weights (or neurons) for previously learned tasks during the training of new tasks. EWC identifies these important weights by calculating the Fisher Information Matrix during or after the training of a task, which estimates how sensitive each weight is to the task's performance. Weights that significantly impact the output for a given task are marked as important. A quadratic penalty is then applied during future learning, constraining these weights to

remain close to their values from the previous task. This ensures that knowledge from earlier tasks is preserved while still allowing the model to adapt to new tasks. However, EWC is **less suitable for LLMs**, which **do not have clearly defined tasks** when it comes to knowledge ingestion (probably different for other types of skills). EWC's effectiveness relies on distinct task boundaries and the ability to compute task-specific importance for weights, which is feasible in scenarios with well-defined tasks, such as classification or reinforcement learning. In LLMs, where learning spans a wide range of topics and linguistic structures without clear task delineation, it's challenging to apply EWC's task-based strategy. The model would struggle to assign specific neurons or weights to individual tasks or concepts, making it difficult to protect task-specific knowledge without hindering the model's overall generalization ability across a diverse dataset.

We cite in the remainder more recent literature that we project onto our taxonomy.

Bai et al. (2024) introduce a novel approach to continual learning that leverages global prototypes to mitigate catastrophic forgetting in neural networks. Their key insight is that maintaining stable connections between task-specific representations and pre-learned, general-purpose token embeddings (which serve as global prototypes) can significantly reduce forgetting without requiring explicit replay mechanisms. Through empirical validation on both task-incremental and class-incremental NLP scenarios, they demonstrate that models preserving strong connections to these global prototypes exhibit enhanced stability. While their work shares our goal of preserving knowledge during updates, it differs fundamentally in its approach and granularity: where they focus on task-level knowledge preservation through architectural mechanisms, our work addresses the more specific challenge of managing contradictory factual updates through cognitive-inspired conflict detection. Their finding that stable reference points aid knowledge retention is conceptually relevant to our work, though our results suggest that such architectural approaches alone may be insufficient when handling explicitly contradictory information, where more sophisticated cognitive mechanisms become necessary.

Benjamin et al. (2024) proposed an elegant theoretical framework that interprets neural networks as Bayesian ensembles of classifiers. Their key insight is that a neural network with N parameters can be viewed as a weighted ensemble of N classifiers in the lazy regime, where the classifiers remain fixed throughout learning. This interpretation reveals that a properly designed posterior update rule, resembling SGD without momentum, can enable continual learning without forgetting - notably, they prove that momentum actually exacerbates forgetting. While their work focuses on preserving all knowledge in task-incremental learning, our paper specifically examines cases where knowledge needs to be deliberately updated or overridden. Their key contribution is showing that catastrophic forgetting is linked to the transition from lazy to rich regimes in neural networks, providing both a theoretical explanation for why larger models are more robust to forgetting and a biologically-inspired mechanism for knowledge preservation that perhaps complements our cognitive-based approach.

Elsayed & Mahmood (2024) propose UPGD (Utility-based Perturbed Gradient Descent), a novel approach targeting both catastrophic forgetting and loss of plasticity in streaming learning scenarios. Their method protects useful network units while maintaining plasticity in less-used ones through utility-gated gradient updates and perturbations. Unlike previous approaches requiring task boundaries or memory buffers, UPGD operates in a challenging streaming setting with continuous non-stationarity. Using their newly introduced direct plasticity metric, they demonstrate UPGD's ability to maintain performance levels that surpass or match existing methods. This work complements our investigation by providing evidence that selective neuronal updates based on utility metrics can effectively balance stability and plasticity, though in a task-learning rather than knowledge-updating context.

Hiratani (2024) analyze how task similarity affects continual learning through a novel theoretical framework combining teacher-student models with latent structure. Their key insight is that high input feature similarity coupled with low readout similarity leads to catastrophic outcomes in both knowledge transfer and retention, even when tasks are positively correlated. They demonstrate that weight regularization in the Fisher information metric robustly helps retention regardless of task similarity, while common approaches like activity gating improve retention at the cost of transfer performance. Their theoretical predictions are validated on permuted MNIST tasks with latent variables.

Jha et al. (2024) propose a probabilistic approach to continual learning for vision-language models, specifically focusing on CLIP adaptation. Their method, CLAP, introduces visual-guided attention and task-specific probabilistic adapters to model the distribution of text features, while leveraging CLIP's pre-trained knowledge for initialization and regularization. This work demonstrates that probabilistic modeling can significantly reduce catastrophic forgetting in class-incremental learning scenarios, achieving state-of-the-art performance across multiple benchmarks.

Jiao et al. (2024) propose VQ-Prompt, a novel prompt-based continual learning framework that addresses class-incremental learning with pretrained vision transformers. Their key innovation is incorporating vector quantization into prompt selection, enabling end-to-end optimization of discrete prompts with task loss while maintaining effective knowledge abstraction. This contrasts with our cognitive-dissonance aware approach, as they focus on task adaptation through prompt engineering rather than explicit conflict detection. Their empirical results on ImageNet-R and CIFAR-100 demonstrate superior performance compared to existing prompt-based methods, suggesting the effectiveness of discrete knowledge representation in continual learning.

Lee et al. (2024) propose IsCiL, a framework for continual imitation learning that uses retrievable skills and adapter-based architecture to enable efficient knowledge sharing across tasks. Unlike traditional approaches that isolate task-specific parameters, IsCiL introduces a prototype-based skill retrieval mechanism that allows selective reuse of previously learned skills for new tasks. While focused primarily on motor skills rather than resolving knowledge contradictions, their empirical results show that this selective adaptation approach significantly improves sample efficiency and reduces catastrophic forgetting compared to other adapter-based methods, particularly in scenarios with incomplete demonstrations.

Lee et al. present a systematic empirical investigation of how dataset bias affects continual learning. Through carefully designed experiments across task-incremental, domain-incremental, and class-incremental scenarios, they reveal that bias transfers both forward and backward between tasks. Their analysis shows that CL methods focusing on stability tend to preserve and propagate biases from previous tasks, while emphasis on plasticity allows new biases to contaminate previous knowledge. Based on these insights, they propose BGS (Balanced Greedy Sampling), a method that mitigates bias transfer by maintaining a balanced exemplar memory and retraining the classification head. Note that here, we used "Replay" for Memory Usage in the table since their best performing method (BGS) uses an exemplar memory, but they also evaluate methods without memory.

Peng et al. (2024) proposed a continual learning approach that automates task selection through vector space retrieval, eliminating the need for explicit task IDs, experience replay, or optimization constraints. Their method, Scalable Language Model (SLM), combines Joint Adaptive Reparameterization with dynamic knowledge retrieval to automatically identify relevant parameters for each input, enabling task-agnostic updates. While achieving state-of-the-art results across diverse tasks and model scales (BERT, T5, LLaMA-2), their key contribution is demonstrating that automatic task identification and parameter selection can enable continual learning without requiring explicit task boundaries or memory buffers.

Seo et al. (2024) presented Train-Attention, an interesting meta-learning approach for continual knowledge learning (CKL) in LLMs that predicts and applies weights to tokens *based on their usefulness for future tasks*. Unlike previous approaches that uniformly update all parameters, their method enables *targeted knowledge updates by learning which tokens are most important* to focus on. Through experiments on LAMA-CKL and TemporalWiki benchmarks, they show that selective token-weighted learning significantly reduces catastrophic forgetting while improving learning speed. The work somewhat complements our cognitive-inspired approach, and demonstrates the benefits of selective attention, but it does not explicitly address the handling of contradictory information.

Wang et al. (2024) proposed a unified framework for continual learning that reveals common mathematical structures across seemingly distinct approaches (regularization-based, Bayesian-based, and memory-replay). Building on this unification, they introduce "refresh learning" - a plug-in mechanism that first unlearns current data before relearning it, inspired by the beneficial role of forgetting in human cognition. Their work primarily focuses on task-incremental and class-incremental scenarios, demonstrating improved accuracy across CIFAR and Tiny-ImageNet benchmarks. While their approach differs from our fact-level knowledge updates in LLMs, their findings about selec-

tive forgetting complement our observations about cognitive-inspired update mechanisms. Their theoretical analysis showing that refresh learning improves the flatness of the loss landscape offers an interesting perspective on how controlled forgetting might benefit knowledge retention in neural networks.

Xu et al. (2024) propose a cross-domain task-agnostic incremental learning framework (X-TAIL) for vision-language models, focusing on the challenge of preserving both incrementally learned knowledge and zero-shot abilities. Their approach, RAIL, uses recursive ridge regression with non-linear projections to adapt to new domains without catastrophic forgetting. Unlike previous work requiring domain identity hints or reference datasets, RAIL can classify images across both seen and unseen domains without domain hints, demonstrating superior performance in both discriminative ability and knowledge preservation. While their work advances the technical aspects of continual learning, it differs from our cognitive-inspired investigation as it doesn't address the fundamental challenge of detecting and resolving conflicting knowledge, instead focusing on domain adaptation without explicit conflict awareness.

Zhao et al. (2024) propose a class-incremental learning framework for pre-trained vision models that balances stability and plasticity through two complementary parameter-efficient tuning mechanisms. Their SAFE approach first inherits generalizability from pre-trained models via a "slow learner" that captures transferable knowledge in the first session, then maintains plasticity through a "fast learner" that continuously adapts to new classes while resisting catastrophic forgetting. While focused on vision tasks rather than language models, their dual-speed learning strategy presents interesting parallels to our cognitive-inspired approach – particularly in how both works identify the importance of selective plasticity and the distinction between stable ("stubborn") and adaptable ("plastic") parameters. However, SAFE doesn't address the fundamental challenge of detecting and handling contradictory information that we identify as crucial for true cognitive-inspired learning.

*Unlike the above work, our goal is to understand the fundamental cognitive mechanisms underlying the continuous knowledge updates in LLMs, particularly focusing on how models can detect and react to contradictory information. Rather than proposing a new continual learning method, we provide crucial insights into how different types of knowledge updates affect model behavior and stability.*

### A.2 KNOWLEDGE EDITING

Next, a big portion of recent literature has focused on understanding and modifying the internal knowledge of Large Language Models (LLMs), post-training. Such knowledge editing aims to alter specific facts or associations within the model without the need for full retraining.

Geva et al. (2020) were among the first to show that transformer Feed-Forward Network (FFN) layers act as unnormalized key-value stores encoding relational knowledge inside LLMs. This observation was later confirmed and complemented by others (Meng et al., 2022a; Dai et al., 2021) before being leveraged by subsequent work to master the editing of internal memories. Meng et al. (2022a) introduced ROME (Rank-One Model Editing), a method that uses causal tracing to empirically locate the layers essential to encoding a given association. They then modify these modules by applying small rank-one changes. To identify the relevant modules, they run the network multiple times, introducing corruptions to the input sequence to disturb the inference, and then restore individual states from the original non-corrupted pass. But this work an others worked only on single edits, and were often evaluated one edit at a time, starting each time from a fresh pre-trained model. The same authors later developed MEMIT, which follows the same causal tracing principle but with the goal of scaling up to 10,000 edits in bulk(Meng et al., 2022c). Similarly, Dai et al. (2021) leveraged the identification of knowledge neurons to perform "knowledge surgery" – editing factual knowledge within Transformers without the need for additional fine-tuning. Zhu et al. (2020) approached the knowledge modification task as a constrained optimization problem. Their work found that constrained layer-wise fine-tuning emerges as an effective method for modifying the knowledge that Transformers learn, suggesting a different pathway for knowledge editing inside LLMs. **?** proposed KNOWLEDGEEDITOR, which achieved knowledge editing by training a hyper-network with constrained optimization to modify specific facts without fine-tuning or changing the overall stored knowledge. The method was demonstrated on smaller models like BERT for

fact-checking and BART for question answering, achieving consistent changes in predictions across different formulations of queries.

Li et al. (2023) empirically investigate the pitfalls of knowledge editing in LLMs, revealing two critical issues: logical inconsistencies between multiple edits (like contradictory relationship updates) and knowledge distortion (where edits irreversibly damage the model's knowledge structure). Through carefully designed benchmarks CONFLICTEDIT and ROUNDEDIT, they demonstrate that current editing methods struggle with these challenges, particularly when handling reverse relationships or composite logical rules. While their work focuses on identifying limitations in maintaining logical consistency across edits, our paper takes a complementary cognitive-inspired perspective by addressing how models handle contradictions with their existing knowledge base. Their findings about knowledge distortion align with and reinforce our observations about the catastrophic nature of updates that modify existing knowledge.

Similarly, Huang et al. (2024) empirically investigate causes of performance degradation during knowledge editing in LLMs. They show degradation correlates with editing target complexity and L1-norm growth in edited layers. Their proposed Dump for Sequence (D4S) method regulates layer norm growth using O(1) space complexity, enabling multiple effective updates while minimizing model degradation. Their work provides valuable insights into the mechanisms of model degradation during knowledge editing, but it does not specifically address the distinction between contradictory and non-contradictory updates, as we do in this paper.

Tan et al. (2023) propose MALMEN, a scalable hypernetwork approach for editing Large Language Models by aggregating parameter shifts using a least-squares formulation. While previous editing methods like MEND (Mitchell et al., 2022) could handle only a few facts simultaneously, MALMEN can efficiently edit thousands of facts while maintaining comparable performance. Their key innovation lies in separating the computation between the hypernetwork and LM, enabling arbitrary batch sizes and reducing memory requirements. Their empirical results show that MALMEN can edit hundreds of times more facts than MEND while maintaining similar performance levels, though they note that the method still struggles with generalizing to rephrasing not seen during training. Like other editing approaches, MALMEN focuses on the mechanics of (by design conflicting) updates.

*Unlike all the work above, our goal in this work is not to edit knowledge, but to understand the fundamental mechanisms and phenomena that govern how LLMs integrate new information with existing knowledge. By taking a cognitive-inspired approach focused on dissonance awareness and adaptive plasticity, we reveal critical insights about the nature of knowledge representation and updating in these models.*

## B  EXTRACTION OF HISTORICAL ACTIVATIONS AND GRADIENTS

We here detail our procedure for the extraction of activations and gradients. Source code is also available at `https://figshare.com/s/81f7108d823b5e08e8ec` for ultimate level of details and reproducibility purposes.

### B.1  PRELIMINARY NOTATION

We focus on the historical tracking of gradients of the outputs (grad_outs) and activations for four key matrices within each block of the transformer model: $\text{Attn}_{\text{c\_attn}}$, $\text{Attn}_{\text{c\_proj}}$, $\text{MLP}_{\text{c\_fc}}$, and $\text{MLP}_{\text{c\_proj}}$.

Given an input sequence $X \in \mathbb{R}^{B \times N \times d_{\text{model}}}$, where $B$ is the batch size, $N$ is the sequence length, and $d_{\text{model}}$ is the model dimension, the transformer block is defined as follows:

**Attention Layer:**  The attention mechanism computes query $Q$, key $K$, and value $V$ matrices:
$$Q = XW_Q, \quad K = XW_K, \quad V = XW_V$$
where $W_Q \in \mathbb{R}^{d_{\text{model}} \times d_{\text{key}}}$, $W_K \in \mathbb{R}^{d_{\text{model}} \times d_{\text{key}}}$, and $W_V \in \mathbb{R}^{d_{\text{model}} \times d_{\text{value}}}$ are trainable projection matrices.

The concatenated matrix $\text{Attn}_{\text{c\_attn}}$ is:
$$\text{Attn}_{\text{c\_attn}} = [Q, K, V] = XW_{\text{attn}}$$

where $W_{\text{attn}} = [W_Q, W_K, W_V] \in \mathbb{R}^{d_{\text{model}} \times (2d_{\text{key}} + d_{\text{value}})}$.

The attention context $\text{Attn}_{\text{context}}$ is computed as:

$$\text{Attn}_{\text{context}} = \text{softmax}\left(\frac{QK^T}{\sqrt{d_{\text{key}}}}\right) V$$

The projected attention output $\text{Attn}_{\text{c\_proj}}$ is:

$$\text{Attn}_{\text{c\_proj}} = \text{Attn}_{\text{context}} W_{\text{proj}}$$

where $W_{\text{proj}} \in \mathbb{R}^{d_{\text{value}} \times d_{\text{model}}}$.

**MLP Layer:** The MLP layer consists of two linear transformations with an activation function $\sigma$:

$$\text{MLP}_{\text{c\_fc}} = \sigma(X W_{\text{fc}} + b_{\text{fc}})$$

where $W_{\text{fc}} \in \mathbb{R}^{d_{\text{model}} \times d_{\text{ff}}}$ and $b_{\text{fc}} \in \mathbb{R}^{d_{\text{ff}}}$.

The projected MLP output $\text{MLP}_{\text{c\_proj}}$ is:

$$\text{MLP}_{\text{c\_proj}} = \text{MLP}_{\text{c\_fc}} W_{\text{proj}} + b_{\text{proj}}$$

where $W_{\text{proj}} \in \mathbb{R}^{d_{\text{ff}} \times d_{\text{model}}}$ and $b_{\text{proj}} \in \mathbb{R}^{d_{\text{model}}}$.

## B.2 HISTORICAL GRADIENT AND ACTIVATION COLLECTION

Collecting a profile of neuron activity during training or simulation of training is needed as (i) input feature to know if a fact is dissonant, novel or known, and (ii) as means to identify where to locate targeted updates.

During training, we collect and cumulate the gradients of the outputs (grad_outs) and activations for the matrices $\text{Attn}_{\text{c\_attn}}$, $\text{Attn}_{\text{c\_proj}}$, $\text{MLP}_{\text{c\_fc}}$, and $\text{MLP}_{\text{c\_proj}}$. Let $t$ denote the training step. We collect activations at step $t$:

$$\text{Attn}_{\text{c\_attn}}(t), \text{Attn}_{\text{c\_proj}}(t), \text{MLP}_{\text{c\_fc}}(t), \text{MLP}_{\text{c\_proj}}(t)$$

as well as Gradient of the Outputs (grad_outs) at step $t$ :

$$\nabla L(\text{Attn}_{\text{c\_attn}}(t)), \nabla L(\text{Attn}_{\text{c\_proj}}(t)), \nabla L(\text{MLP}_{\text{c\_fc}}(t)), \nabla L(\text{MLP}_{\text{c\_proj}}(t))$$

In the remainder, we denote these, regardless of their provenance matrix, as:

$$A^l(t), G^l(t) \in \mathbb{R}^{B \times N \times d_{\text{out}}^l}$$

where $l$ denotes the layer, $B$ is the batch size, $N$ is the sequence length, and $d_{\text{out}}^l$ is the output dimension of layer $l$.

When needed, we standardize these metrics for each layer $l$ as follows:

$$\hat{A}^l(t) = \frac{A^l(t) - \mu_A^l(t)}{\sigma_A^l(t)}, \quad \hat{G}^l(t) = \frac{G^l(t) - \mu_G^l(t)}{\sigma_G^l(t)}$$

where $\mu$ and $\sigma$ are the mean and standard deviation computed over all dimensions of the respective tensor.

We then sum over the batch dimension:

$$S_{\hat{A}}^l(t)_{n,i} = \sum_{b=1}^{B} \hat{A}_{b,n,i}^l(t), \quad S_{\hat{G}}^l(t)_{n,i} = \sum_{b=1}^{B} \hat{G}_{b,n,i}^l(t)$$

Optionally[5], we can sum over the token dimension:

---

[5]We consider two approaches. In the first, we extract the activations and gradients corresponding to the last token (i.e., position $N$) in the sequence for each sample in the batch. This is reasonable since the last token is representative of the fact or information of interest in our datasets. In the second, we simply aggregate over all tokens, where we aggregate activations and gradients across all tokens in the sequence by computing statistical measures such as the mean or sum over the token dimension.

$$S_{\hat{A}}^l(t)_i = \sum_{n=1}^{N} S_{\hat{A}}^l(t)_{n,i}, \quad S_{\hat{G}}^l(t)_i = \sum_{n=1}^{N} S_{\hat{G}}^l(t)_{n,i}$$

The standardized and summed metrics are then accumulated across the training steps:

$$H\hat{A}_i^l = \sum_{t=1}^{T} S_{\hat{A}}^l(t)_i, \quad H\hat{G}_i^l = \sum_{t=1}^{T} S_{\hat{G}}^l(t)_i$$

where $T$ is the total number of training steps.

These historical activations $H\hat{A}^l$ and gradients $H\hat{G}^l$ provide cumulative measures of neuron activity over the training process. They help identify neurons that are heavily utilized (stubborn neurons) and those that are underutilized (plastic neurons), which is crucial for our targeted updates.

## C    DISSONANCE AWARENESS

### C.1    AUGMENTING THE COUNTERFACT DATASET WITH NOVEL FACTS

To generate unknown facts to augment the Counterfact dataset, we used GPT-3.5 with a prompt as follows:

```
Starting from this list of facts, can you create one data entry for each
    that concerns imaginary names and characters if necessary, while
    following the same logic.

For example, Danielle Darrieux's mother tongue is French => Becomes
    Machin De Machine's mother tongue is Kurdi (or Kinduli).

Edwin of Northumbria's religious values strongly emphasize Christianity
    => Hamed Habib's religious values strongly emphasize Atheism (or
    Peace or..)

Try to make the old and new as far as possible from each other (e.g.,
    Kurdi is far from French, Kinduli is an imaginary language, etc.),
    while keeping some logic.

Write in JSON format, please (easy to parse):

- Danielle Darrieux's mother tongue is French
- Edwin of Northumbria's religious values strongly emphasize Christianity
- Toko Yasuda produces the most amazing music on the guitar
- One can get to Autonomous University of Madrid by navigating Spain
- Thomas Joannes Stieltjes was born in Dutch
- Anaal Nathrakh originated from Birmingham
```

**Example Generated Transformations:**

- Original: *"Toko Yasuda produces the most amazing music on the guitar."*
  Transformed: *"Zara Zorin produces the most amazing music on the theremin."*

- Original: *"One can get to Autonomous University of Madrid by navigating Spain."*
  Transformed: *"One can reach the Floating Academia of Zephyria by navigating through the Cloud Realms."*

- Original: *"Thomas Joannes Stieltjes was born in Dutch."*
  Transformed: *"Lorien Ilithar was born amidst the Elvish."*

These transformations help create novel facts unlikely to be known by the model, enabling us to evaluate its ability to handle unknown information effectively.

Table 5: *Ablation study of dissonance awareness:* Classification Results for Different Scenarios, Feature Sets, Normalization strategies and Classifier. Average (and std) accuracy and F1 scores. ⋆ denotes the best combination for each classifier

| Scenario | Features | Normalization | Classifier | Accuracy | F1 Score |
|---|---|---|---|---|---|
| Finetuned | A+G | Null | SVM | 0.994 (0.004) | 0.994 (0.004) |
| | | | RF⋆ | 0.988 (0.001) | 0.988 (0.001) |
| | | Layer | SVM | 0.995 (0.001) | 0.995 (0.001) |
| | | | RF | 0.982 (0.005) | 0.982 (0.004) |
| | | Historical | SVM⋆ | 0.995 (0.001) | 0.995 (0.001) |
| | | | RF | 0.978 (0.003) | 0.978 (0.003) |
| | G | Null | SVM | 0.917 (0.009) | 0.918 (0.009) |
| | | | RF | 0.905 (0.008) | 0.906 (0.008) |
| | | Layer | SVM | 0.920 (0.003) | 0.921 (0.003) |
| | | | RF | 0.895 (0.007) | 0.896 (0.007) |
| | | Historical | SVM | 0.897 (0.004) | 0.898 (0.004) |
| | | | RF | 0.868 (0.014) | 0.870 (0.014) |
| | A | Null | SVM | 0.796 (0.005) | 0.796 (0.007) |
| | | | RF | 0.747 (0.012) | 0.745 (0.016) |
| | | Layer | SVM | 0.783 (0.013) | 0.784 (0.012) |
| | | | RF | 0.722 (0.009) | 0.720 (0.007) |
| | | Historical | SVM | 0.781 (0.009) | 0.781 (0.010) |
| | | | RF | 0.721 (0.010) | 0.719 (0.008) |
| Pretrained | A+G | Null | SVM | 0.944 (0.006) | 0.944 (0.006) |
| | | | RF⋆ | 0.928 (0.012) | 0.929 (0.011) |
| | | Layer | SVM | 0.949 (0.006) | 0.949 (0.006) |
| | | | RF | 0.909 (0.014) | 0.910 (0.013) |
| | | Historical | SVM⋆ | 0.947 (0.004) | 0.948 (0.003) |
| | | | RF | 0.925 (0.006) | 0.925 (0.006) |
| | G | Null | SVM | 0.904 (0.006) | 0.904 (0.006) |
| | | | RF | 0.891 (0.010) | 0.892 (0.009) |
| | | Layer | SVM | 0.902 (0.008) | 0.902 (0.007) |
| | | | RF | 0.859 (0.013) | 0.861 (0.011) |
| | | Historical | SVM | 0.915 (0.007) | 0.916 (0.006) |
| | | | RF | 0.879 (0.017) | 0.879 (0.016) |
| | A | Null | SVM | 0.909 (0.006) | 0.909 (0.006) |
| | | | RF | 0.894 (0.009) | 0.895 (0.007) |
| | | Layer | SVM | 0.905 (0.012) | 0.905 (0.011) |
| | | | RF | 0.876 (0.004) | 0.877 (0.003) |
| | | Historical | SVM | 0.900 (0.008) | 0.900 (0.007) |
| | | | RF | 0.881 (0.006) | 0.882 (0.006) |

## C.2 ABLATION STUDY OF CLASSIFIER PERFORMANCE

We conducted an extended ablation study of the dissonance awareness classifier, evaluating its performance under different scenarios (fine-tuned vs. pre-trained models), feature sets (A, G, A+G), normalization strategies (None, Layer, Historical), and classifiers (Random Forests (RF) and Support Vector Machines (SVM)).

Table 5 presents a comprehensive set of classification results, including average accuracy and F1 scores (with standard deviations) across different settings. The best results for each classifier are denoted with a ⋆ and reported earlier in Table 2 in the main paper.

## C.3 EXPLANATION OF FEATURE IMPORTANCE

To further understand the discriminative power of different features, we analyzed the feature importance scores derived from the RF classifier.

First, as earlier mentioned in Fig.3 in the main paper, gradient-based features are substantially more important than activation-based features. This suggests that fine-tuning leads to more discriminative gradients, possibly due to the model overfitting on the known facts, resulting in near-zero gradients for known facts and higher gradients for novel or conflicting facts. In contrast, for the pre-trained model, both activation and gradient features contribute significantly, indicating that combining internal representations and learning dynamics is beneficial for classification.

Complementary to Fig.3, block importance reported in Fig. 8 reveals that, in the pre-trained model all transformer blocks tend to contribute relatively equally to the classification task, with the last layers contributing less. The finetuned model, on the other hand shows a slightly different tendency where the earlier layers contribute less. More work is clearly needed to understand such differences. This paper focuses only on feasibility of the entire cognitive-dissonance approach, leaving more elaborate evaluations for future work.

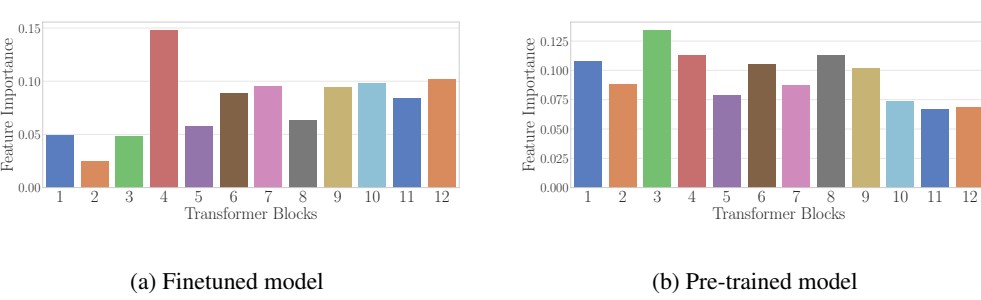

(a) Finetuned model  (b) Pre-trained model

Figure 8: *Block Importance.* Albeit differences are visible, the tendency is not as marked as for the activation vs gradient based feature importance in Fig.3 - GPT2-small

### C.4 LOCATION OF STUBBORN NEURONS

We also report the distribution of stubborn neurons across the transformer blocks in GPT-2 XL. Figures 9a and 9b show histograms of the number of stubborn neurons identified in each block for thresholds of 8,000 and 2,000 neurons, respectively.

Our analysis indicates that stubborn neurons are not uniformly distributed throughout the network. Instead, they curiousy tend to be concentrated in certain blocks, particularly in the first block and in certain middle layers of the transformer. This might suggest that these layers play a more significant role in encoding and retaining knowledge during training. Interestingly, $\text{Attn}_{c\_attn}$ concentrates much more of the stubborn neurons overall, with the exception of the first block where $\text{Attn}_{c\_proj}$ has a substantially higher share of stubborn neurons. The results are similar for both thresholds.

Overall, understanding the distribution of stubborn neurons can inform targeted update strategies by identifying which parts of the network are more critical for preserving existing knowledge.

### C.5 ALTERNATIVE FEATURES FOR DISSONANCE AWARENESS

In this work, we used activations and gradients as they were *readily available* in our experimental pipeline. We now test whether using model output only, which is more easily available than internal gradients and activations can achieve similar performance on our scenario.

Each fact in our dataset is conceptually a statement involving a subject (s), relation (r), and object (o) (e.g., "Danielle Darrieux's mother tongue is French"). In this section, we extract features that capture increasing levels of detail about the model's predictions, related to what the actual facts are, leveraging both:

- Conditional probabilities $p(o|s, r)$ at different truncation points[6]

---

[6]Since the object $o$ can span multiple tokens, we extract features from the last $N$ tokens of each fact (we pick three, since most answers fit within that limit). For each token position, we compute both the truncated prompt probability $p(o|s, r)$ by removing the token and subsequent tokens, and the full sentence probability

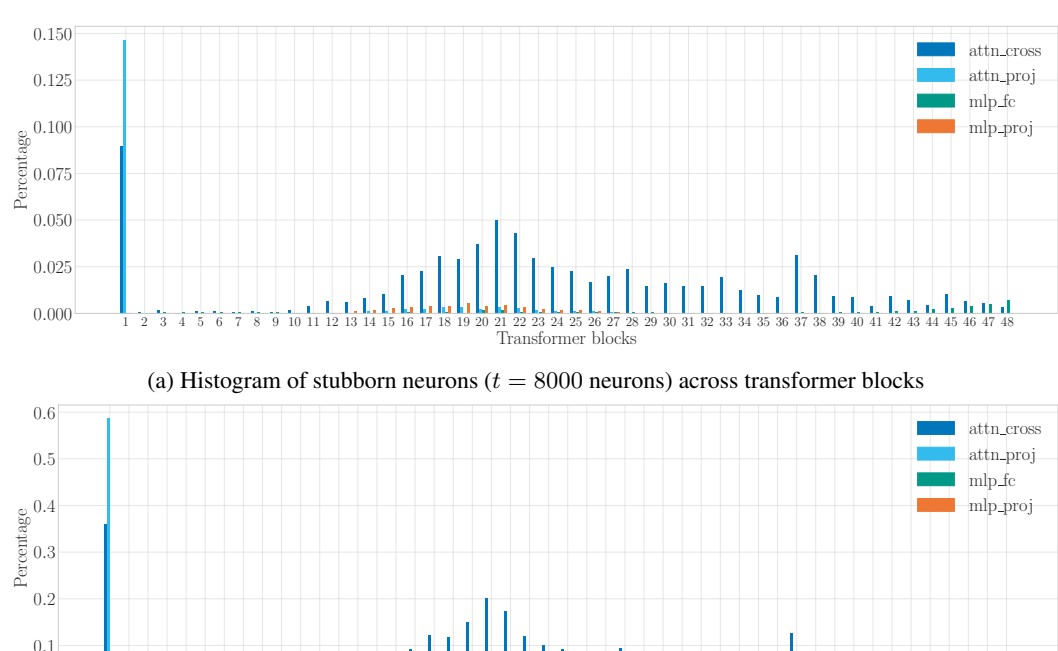

(a) Histogram of stubborn neurons ($t = 8000$ neurons) across transformer blocks

(b) Histogram of stubborn neurons ($t = 2000$ neurons) across transformer blocks

Figure 9: Distribution of stubborn neurons across gpt2-xl transformer blocks for different neuron thresholds to define stubbornness. (a) shows the distribution for $t = 8000$ neurons, while (b) corresponds to $t = 2000$ neurons.

- Joint probability $p(s, r, o)$ of the full statement

In more details, we extract the following features, with increasing complexity.

**Basic Token Probabilities** ($Feat_1$)**:** For each of the last $N$ tokens (representing the answer), we collect the probability of the actual next token given the truncated prompt. These simple scalar features capture the model's direct confidence in the correct continuation. This has a dimensionality of $N + 1$ ($N$ truncation points plus full statement, so 4 in our case.)

**Top-$k$ Predictions Analysis** ($Feat_2$)**:** Here, for each position in the answer, we collect the values and normalized indices of top-$k$ most likely next tokens. This captures both confidence distribution and ranking patterns. Similarly to the above, we compute this for both truncated prompts and full statements. Here, the dimensionality is $(N + 1) \times 2k$ ($k$ values and $k$ normalized indices for each position). We pick k=100.

**Distribution Features** ($Feat_3$)**:** Here, we analyze the complete probability distribution over the vocabulary. For each position in the answer sequence, we construct histograms of the probabilities with $n_{bins}$ bins (here 100), capturing the full spectrum of the model's prediction patterns. We augment these distributions with indicator vectors that highlight the positions of ground truth tokens (the true next tokens of the current truncated fact), providing additional context about the model's accuracy. This results in a feature vector of dimensionality $(N + 1) \times n_{bins}$.

**Combined Features** ($Concat$)**:** Here, we simply concatenate $Feat_1$, $Feat_2$, and $Feat_3$.

Tab. 6 shows the results over our dataset. We observe a similar great performance when using the model outputs, compared to Activations and Gradients. Model output achieves even better performance in case of pre-trained models. This is inline with our earlier observation that activations (what

---

$p(s, r, o)$. This multi-token analysis ensures we capture the model's predictions across the entire span of the answer.

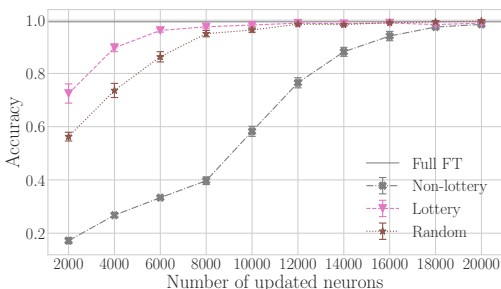

Figure 10: Lottery ticket

we're using now) are more important than gradients in the case of pre-trained models. This result is encouraging for future work, where we plan to (i) build more challenging classification datasets (than the simple facts in CounterFact) and (ii) build standalone classifiers to speed up the training of LLMs, by avoiding training on conflicting data.

| Strategy (dim) | Pretrained Model | | Finetuned Model | |
|---|---|---|---|---|
| | Accuracy | F1-Score | Accuracy | F1-Score |
| Feat.1 (4) | 0.852 | 0.856 | 0.850 | 0.855 |
| Feat.2 (800) | 0.602 | 0.588 | 0.600 | 0.581 |
| Feat.3 (400) | 0.540 | 0.452 | 0.543 | 0.464 |
| Concat (1204) | 0.983 | 0.983 | 0.978 | 0.978 |
| (A+G) (240) | 0.947 | 0.948 | 0.995 | 0.995 |

Table 6: Using output-only features for dissonance-awareness can achieve similar good performance to using our readily available activations and gradients, and even better in the case of the pre-trained model.

# D  NON-DISSONANT UPDATES

## D.1  SIMILARITIES WITH LOTTERY TICKET

To assess the hypothesis that certain subnetworks within the language model are more conducive to integrating new information—a notion earlier named the lottery ticket hypothesis (Frankle & Carbin, 2018)—we designed an experiment to confirm this effect.

We first trained a model on 10,000 disjoint facts (referred to as Facts H) and identified the most active candidate neurons during this process, which we term *Lottery Ticket Neurons*. These neurons should form a preferred subnetwork for representing Facts H. Next, we started from a *fresh model* and trained on a new set of novel facts (Facts A), which are different from H, restricting updates to three distinct groups of neurons:

1. **Lottery Ticket Neurons**: Neurons highly active during the initial training on Facts H.
2. **Non-Lottery Neurons**: Neurons underutilized during the initial training on Facts H.
3. **Random Neurons**: Neurons selected randomly from the entire network.

Figure 10 shows the accuracy of acquiring new knowledge when using each of these strategies, with the number of neurons varying from 2,000 to 20,000. Using the Lottery Ticket Neurons led to significantly better performance, reaching nearly 100% accuracy at 8,000 neurons, compared to around 40% for the Non-Lottery Neurons. The Random Neurons strategy also performed relatively well, interestingly suggesting that capturing even a few "anchor" neurons from the preferred subnetwork is sufficient to achieve good performance.

These results support the existence of preferred subnetworks within the model that are particularly effective for learning new information. Leveraging these subnetworks can enhance the efficiency

of knowledge integration while preserving existing knowledge, an aspect that our candidate and specific strategies are already exploiting.

## D.2 Hyperparameter selection: learning rate and batch size for GPT2-XL

We conducted a hyperparameter search to determine the optimal learning rate and batch size for fine-tuning GPT-2 XL on our dataset. Table 7 presents the performance of the model on old and new knowledge across various learning rates and batch sizes.

| Learning Rate | Batch Size | Epochs | Accuracy |
|---|---|---|---|
| 1e-06 | 64 | 5 | 0.271 |
| 1e-06 | 64 | 10 | 0.476 |
| 1e-06 | 64 | 20 | 0.694 |
| 1e-06 | 32 | 5 | 0.441 |
| 1e-06 | 32 | 10 | 0.641 |
| 1e-06 | 32 | 20 | 0.888 |
| 1e-06 | 16 | 5 | 0.582 |
| 1e-06 | 16 | 10 | 0.782 |
| 1e-06 | 16 | 20 | 0.984 |
| **1e-05** | **32** | **5** | **0.981** |
| 1e-05 | 32 | 7 | 0.997 |
| 1e-05 | 16 | 5 | 0.989 |
| 1e-05 | 16 | 7 | 0.997 |
| 1e-05 | 16 | 10 | 0.998 |
| 5e-06 | 32 | 5 | 0.853 |
| 5e-06 | 32 | 7 | 0.957 |
| 5e-06 | 32 | 10 | 0.996 |
| 5e-06 | 16 | 5 | 0.954 |
| 5e-06 | 16 | 7 | 0.996 |
| 5e-06 | 16 | 10 | 0.998 |

Table 7: Accuracy results for different learning rates, batch sizes, and epochs on 10k facts (GPT2-xl). We use the finetuning on 10k facts as a proxy to pick the hyperparameters of our later continual update experiments (learning rate, batch size and epochs). In bold, what we picked for GPT2-xl. Not shown here, for GPT2-small, we picked 5e-4.

## D.3 Comprehensive Analysis of GPT2-XL non-dissonant Updates

Figure 11 presents the accuracy of GPT-2 XL on old and new knowledge under various neuron update strategies and experimental conditions. We explored different configurations to understand how the model's larger capacity affects knowledge integration.

Our results reveal distinct scaling behaviors compared to GPT-2 small. When using the same learning rate as GPT-2 small (Figures 11a, 11b), the model maintains old knowledge but struggles to effectively integrate new information. With the optimal learning rate for GPT-2 XL (Figures 11c, 11d), we observe improved new knowledge acquisition while still preserving old knowledge, though less effectively than with the lower learning rate.

Increasing the learning rate by 10x (Figures 11e, 11f) or allocating 10x more neurons (Figures 11g, 11h) shows that GPT-2 XL requires either higher learning rates or more extensive parameter updates compared to GPT-2 small to achieve effective learning. This suggests that targeted strategies using fewer neurons need to compensate through these adjustments.

Extended training duration (50 epochs, Figures 11i, 11j) allows the model to better integrate new knowledge while preserving old information, indicating that longer training can help overcome the limitations of sparse updates in larger models. Figure 12 summarizes these trade-offs across all configurations, highlighting how different hyperparameter choices affect the balance between preserving old knowledge and acquiring new information.

While GPT-2 XL's larger capacity naturally reduces interference with our tracked facts during non-dissonant updates, this improved performance is "deceptive" and should be interpreted cautiously: *we cannot measure potential effects on other pre-trained knowledge beyond our tracked facts.*

*These results highlight the methodological challenges in studying knowledge updates in larger models: their increased capacity can mask interference with tracked facts, making it harder to fully measure the impact of updates on the model's broader knowledge.* This underscores the importance of controlled experimental settings when studying fundamental properties of knowledge updating in neural networks.

## E    DISSONANT UPDATES

### E.1    IMPACT OF NUMBER OF CONFLICTING FACTS

We examined the effect of varying the number of conflicting facts introduced during dissonant updates. Figure 13 shows the performance metrics of GPT-2 small when editing 10, 100, and 1,000 facts, respectively.

Our findings show that as the number of conflicting facts increases, the impact on old knowledge retention becomes more pronounced, with all strategies experiencing significant degradation. The ability to learn new conflicting knowledge improves slightly with more facts, but overall performance remains suboptimal. The plastic and random neuron strategies tend to preserve old knowledge when editing a small number of facts (e.g., 10 facts), but their effectiveness diminishes as more conflicting information is introduced. Interestingly, the opposite effect is observed for new knowledge, where adding more facts seems to make it easier to learn new knowledge, for all strategies.

### E.2    DETAILED FIGURES FOR SPECIFIC NUMBERS OF NEURONS

Tables 8, Figs. 9, 10, and 11 provide detailed performance metrics for different neuron thresholds (20,000, 6,000, and 4,000 neurons, respectively) when editing 1,000, 100 and 10, conflicting facts using various strategies.

Table 8: Neuron Editing Results for N=20,000 Neurons

| Samples | Strategy | Accuracy A | Accuracy NOT(B) | Accuracy GEN | Harmonic Mean |
|---|---|---|---|---|---|
| 10 | Full Finetune | 0.107 (0.082) | 1.000 (0.000) | 0.576 (0.117) | 0.222 (0.116) |
| | Specific | 0.491 (0.137) | 1.000 (0.000) | 0.604 (0.126) | 0.621 (0.109) |
| | Plastic | 0.735 (0.105) | 0.752 (0.175) | 0.220 (0.183) | 0.434 (0.185) |
| | Stubborn | 0.449 (0.109) | 1.000 (0.000) | 0.616 (0.091) | 0.606 (0.084) |
| | Candidate | 0.430 (0.134) | 1.000 (0.000) | 0.656 (0.125) | 0.597 (0.116) |
| | Random | 0.688 (0.107) | 0.944 (0.083) | 0.448 (0.212) | 0.579 (0.222) |
| 100 | Full Finetune | 0.238 (0.019) | 0.998 (0.003) | 0.434 (0.089) | 0.398 (0.041) |
| | Specific | 0.412 (0.046) | 0.988 (0.005) | 0.330 (0.054) | 0.460 (0.046) |
| | Plastic | 0.317 (0.052) | 0.586 (0.048) | 0.128 (0.028) | 0.233 (0.035) |
| | Stubborn | 0.435 (0.043) | 0.999 (0.002) | 0.427 (0.085) | 0.528 (0.057) |
| | Candidate | 0.463 (0.032) | 0.999 (0.002) | 0.447 (0.083) | 0.552 (0.052) |
| | Random | 0.474 (0.035) | 0.874 (0.048) | 0.292 (0.048) | 0.444 (0.036) |
| 1000 | Full Finetune | 0.182 (0.007) | 0.991 (0.009) | 0.442 (0.053) | 0.341 (0.016) |
| | Specific | 0.188 (0.033) | 0.995 (0.002) | 0.257 (0.025) | 0.292 (0.035) |
| | Plastic | 0.077 (0.021) | 0.996 (0.002) | 0.224 (0.018) | 0.160 (0.027) |
| | Stubborn | 0.185 (0.010) | 0.992 (0.005) | 0.327 (0.013) | 0.317 (0.012) |
| | Candidate | 0.172 (0.018) | 0.996 (0.001) | 0.369 (0.043) | 0.314 (0.028) |
| | Random | 0.235 (0.029) | 0.995 (0.003) | 0.300 (0.053) | 0.347 (0.041) |

The results show that changing the number of neurons allocated for updates does not necessarily improve or degrade performance in the dissonant update scenario. In all cases, the model struggles to retain old knowledge while learning new conflicting information. The candidate and specific neuron strategies are consistently and significantly better than state of the art solutions, offering a

Table 9: Neuron Editing Results for N=8,000 Neurons

| Samples | Strategy | Accuracy A | Accuracy NOT(B) | Accuracy GEN | Harmonic Mean |
|---|---|---|---|---|---|
| 10 | Full Finetune | 0.107 (0.082) | 1.000 (0.000) | 0.576 (0.117) | 0.222 (0.116) |
| | Specific | 0.638 (0.138) | 0.964 (0.039) | 0.512 (0.238) | 0.600 (0.183) |
| | Plastic | 0.909 (0.039) | 0.020 (0.040) | 0.000 (0.000) | 0.0 |
| | Stubborn | 0.622 (0.110) | 0.972 (0.030) | 0.544 (0.169) | 0.643 (0.103) |
| | Candidate | 0.596 (0.106) | 0.988 (0.024) | 0.644 (0.128) | 0.690 (0.058) |
| | Random | 0.827 (0.083) | 0.380 (0.132) | 0.092 (0.094) | 0.277 (0.098) |
| 100 | Full Finetune | 0.238 (0.019) | 0.998 (0.003) | 0.434 (0.089) | 0.398 (0.041) |
| | Specific | 0.531 (0.030) | 0.760 (0.063) | 0.263 (0.027) | 0.426 (0.024) |
| | Plastic | 0.433 (0.029) | 0.059 (0.014) | 0.028 (0.017) | 0.052 (0.025) |
| | Stubborn | 0.530 (0.054) | 0.936 (0.048) | 0.398 (0.064) | 0.547 (0.063) |
| | Candidate | 0.542 (0.035) | 0.969 (0.033) | 0.462 (0.081) | 0.591 (0.054) |
| | Random | 0.508 (0.019) | 0.193 (0.038) | 0.065 (0.025) | 0.131 (0.039) |
| 1000 | Full Finetune | 0.182 (0.007) | 0.991 (0.009) | 0.442 (0.053) | 0.341 (0.016) |
| | Specific | 0.240 (0.017) | 0.993 (0.003) | 0.287 (0.039) | 0.345 (0.028) |
| | Plastic | 0.218 (0.024) | 0.283 (0.026) | 0.070 (0.010) | 0.133 (0.013) |
| | Stubborn | 0.200 (0.007) | 0.995 (0.001) | 0.317 (0.024) | 0.327 (0.006) |
| | Candidate | 0.199 (0.014) | 0.996 (0.002) | 0.380 (0.041) | 0.345 (0.014) |
| | Random | 0.159 (0.032) | 0.784 (0.091) | 0.102 (0.014) | 0.169 (0.010) |

Table 10: Neuron Editing Results for N=6,000 Neurons

| Samples | Strategy | Accuracy A | Accuracy NOT(B) | Accuracy GEN | Harmonic Mean |
|---|---|---|---|---|---|
| 10 | Full Finetune | 0.107 (0.082) | 1.000 (0.000) | 0.576 (0.117) | 0.222 (0.116) |
| | Specific | 0.663 (0.117) | 0.800 (0.111) | 0.436 (0.204) | 0.545 (0.164) |
| | Plastic | 0.941 (0.031) | 0.004 (0.008) | 0.000 (0.000) | 0.0 |
| | Stubborn | 0.641 (0.083) | 0.868 (0.057) | 0.404 (0.160) | 0.548 (0.111) |
| | Candidate | 0.604 (0.115) | 0.956 (0.043) | 0.552 (0.134) | 0.642 (0.057) |
| | Random | 0.898 (0.059) | 0.120 (0.126) | 0.000 (0.000) | 0.0 |
| 100 | Full Finetune | 0.238 (0.019) | 0.998 (0.003) | 0.434 (0.089) | 0.398 (0.041) |
| | Specific | 0.552 (0.014) | 0.573 (0.064) | 0.200 (0.025) | 0.347 (0.020) |
| | Plastic | 0.627 (0.051) | 0.010 (0.005) | 0.011 (0.011) | 0.020 (0.004) |
| | Stubborn | 0.558 (0.050) | 0.850 (0.091) | 0.371 (0.063) | 0.527 (0.062) |
| | Candidate | 0.569 (0.031) | 0.925 (0.091) | 0.436 (0.095) | 0.580 (0.075) |
| | Random | 0.497 (0.047) | 0.077 (0.029) | 0.040 (0.030) | 0.071 (0.041) |
| 1000 | Full Finetune | 0.182 (0.007) | 0.991 (0.009) | 0.442 (0.053) | 0.341 (0.016) |
| | Specific | 0.230 (0.012) | 0.992 (0.006) | 0.297 (0.052) | 0.342 (0.030) |
| | Plastic | 0.270 (0.054) | 0.196 (0.022) | 0.057 (0.010) | 0.112 (0.014) |
| | Stubborn | 0.200 (0.018) | 0.993 (0.005) | 0.315 (0.043) | 0.325 (0.029) |
| | Candidate | 0.185 (0.026) | 0.997 (0.002) | 0.357 (0.048) | 0.322 (0.026) |
| | Random | 0.194 (0.026) | 0.663 (0.072) | 0.088 (0.008) | 0.165 (0.014) |

slight advantage. However, they are still unable to effectively mitigate the destructive effects of dissonant updates, further motivating the neeed for both (i) dissonance awareness and (ii) proper conflict resolution.

### E.3    SCALING TO GPT2-XL

We extended our dissonant update experiments to GPT-2 XL to examine whether our observations about knowledge conflicts persist in larger models.

Figure 14 examines gpt2-xl's behavior when updating 1,000 conflicting facts using the optimal learning rate, as determined by our hyperparameter search. We compare three configurations: GPT-2

Table 11: Neuron Editing Results for N=4,000 Neurons

| Samples | Strategy | Accuracy A | Accuracy NOT(B) | Accuracy GEN | Harmonic Mean |
|---|---|---|---|---|---|
| 10 | Full Finetune | 0.107 (0.082) | 1.000 (0.000) | 0.576 (0.117) | 0.222 (0.116) |
| | Specific | 0.673 (0.101) | 0.656 (0.168) | 0.264 (0.208) | 0.385 (0.182) |
| | Plastic | 0.965 (0.021) | 0.000 (0.000) | 0.000 (0.000) | 0.0 |
| | Stubborn | 0.635 (0.062) | 0.764 (0.087) | 0.352 (0.115) | 0.506 (0.101) |
| | Candidate | 0.603 (0.101) | 0.864 (0.126) | 0.512 (0.106) | 0.613 (0.065) |
| | Random | 0.863 (0.066) | 0.144 (0.113) | 0.044 (0.062) | 0.169 (0.050) |
| 100 | Full Finetune | 0.238 (0.019) | 0.998 (0.003) | 0.434 (0.089) | 0.398 (0.041) |
| | Specific | 0.553 (0.023) | 0.408 (0.040) | 0.137 (0.022) | 0.258 (0.029) |
| | Plastic | 0.760 (0.054) | 0.000 (0.000) | 0.003 (0.003) | 0.0 |
| | Stubborn | 0.565 (0.060) | 0.705 (0.143) | 0.303 (0.077) | 0.460 (0.092) |
| | Candidate | 0.573 (0.041) | 0.852 (0.124) | 0.400 (0.102) | 0.548 (0.093) |
| | Random | 0.487 (0.043) | 0.090 (0.018) | 0.045 (0.023) | 0.082 (0.030) |
| 1000 | Full Finetune | 0.182 (0.007) | 0.991 (0.009) | 0.442 (0.053) | 0.341 (0.016) |
| | Specific | 0.235 (0.008) | 0.976 (0.012) | 0.265 (0.041) | 0.329 (0.025) |
| | Plastic | 0.348 (0.049) | 0.125 (0.021) | 0.047 (0.006) | 0.093 (0.009) |
| | Stubborn | 0.203 (0.013) | 0.989 (0.006) | 0.315 (0.031) | 0.329 (0.016) |
| | Candidate | 0.184 (0.013) | 0.996 (0.001) | 0.370 (0.045) | 0.327 (0.025) |
| | Random | 0.254 (0.049) | 0.400 (0.085) | 0.072 (0.006) | 0.146 (0.010) |

small (2,000 to 20,000 neurons) shown previously, gpt2-xl with the same range, and gpt2-xl with ten times more neurons (20,000 to 200,000). The latter was shown effective in packing new knowledge compared to (2000 to 20000) range in non-dissonant updates.

First, while gpt2-xl still requires more neurons than GPT-2 small to effectively learn new conflicting knowledge, as seen earlier, the key finding concerns old knowledge retention: regardless of model size or neuron allocation, we observe significant degradation of old, unrelated knowledge across all strategies.

Interestingly, this degradation persists even when using fewer neurons and when the model fails to effectively learn the new conflicting information (2k to 20k). These results strongly suggest that the destructive impact of conflicting updates on existing knowledge is a fundamental property that remains present in larger models.

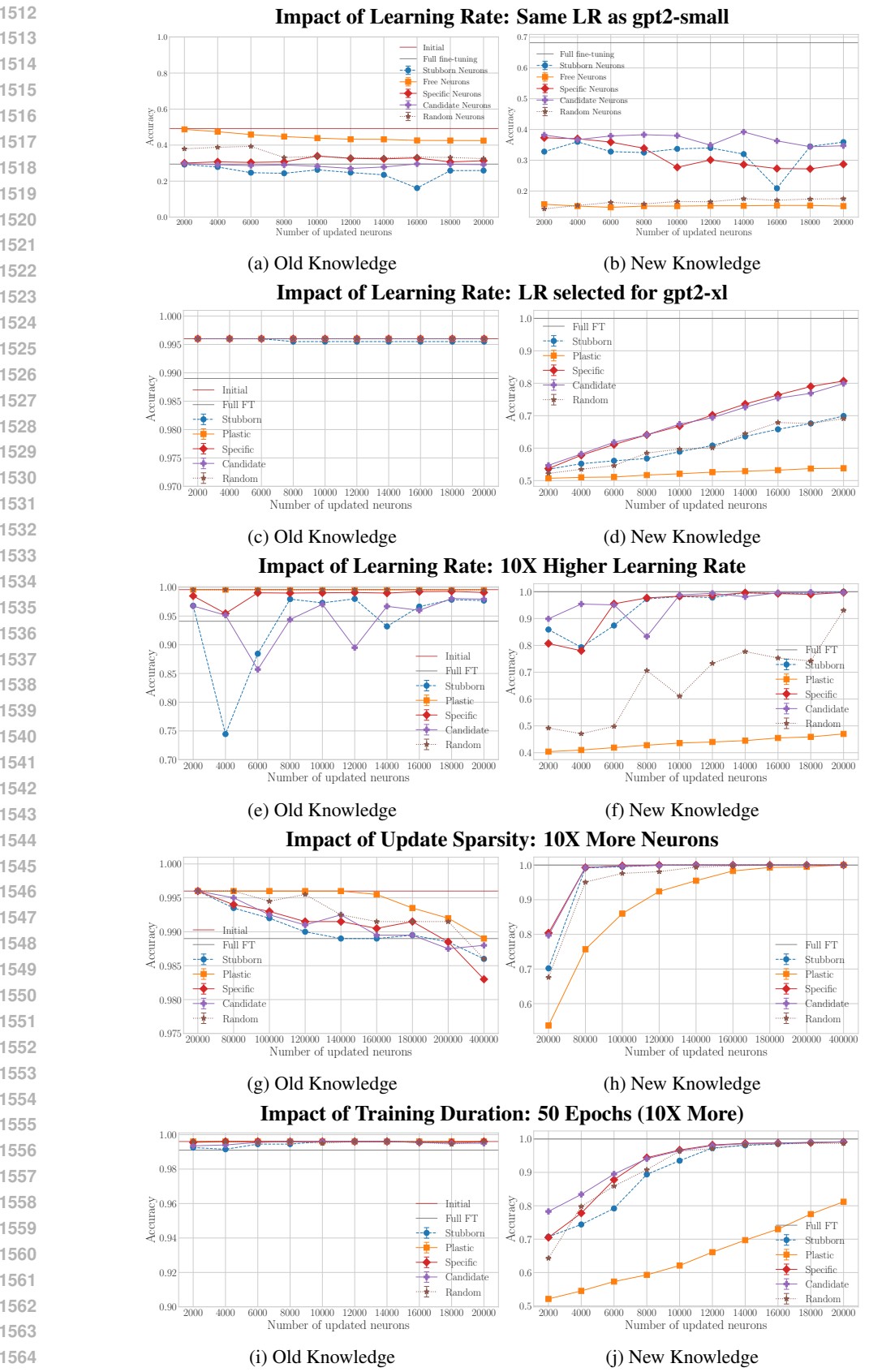

Figure 11: **Non-dissonant update.** gpt2-xl under various conditions. Each row corresponds to a different experimental condition, with the left column showing Old Knowledge and the right column showing New Knowledge. Results on a single fold.

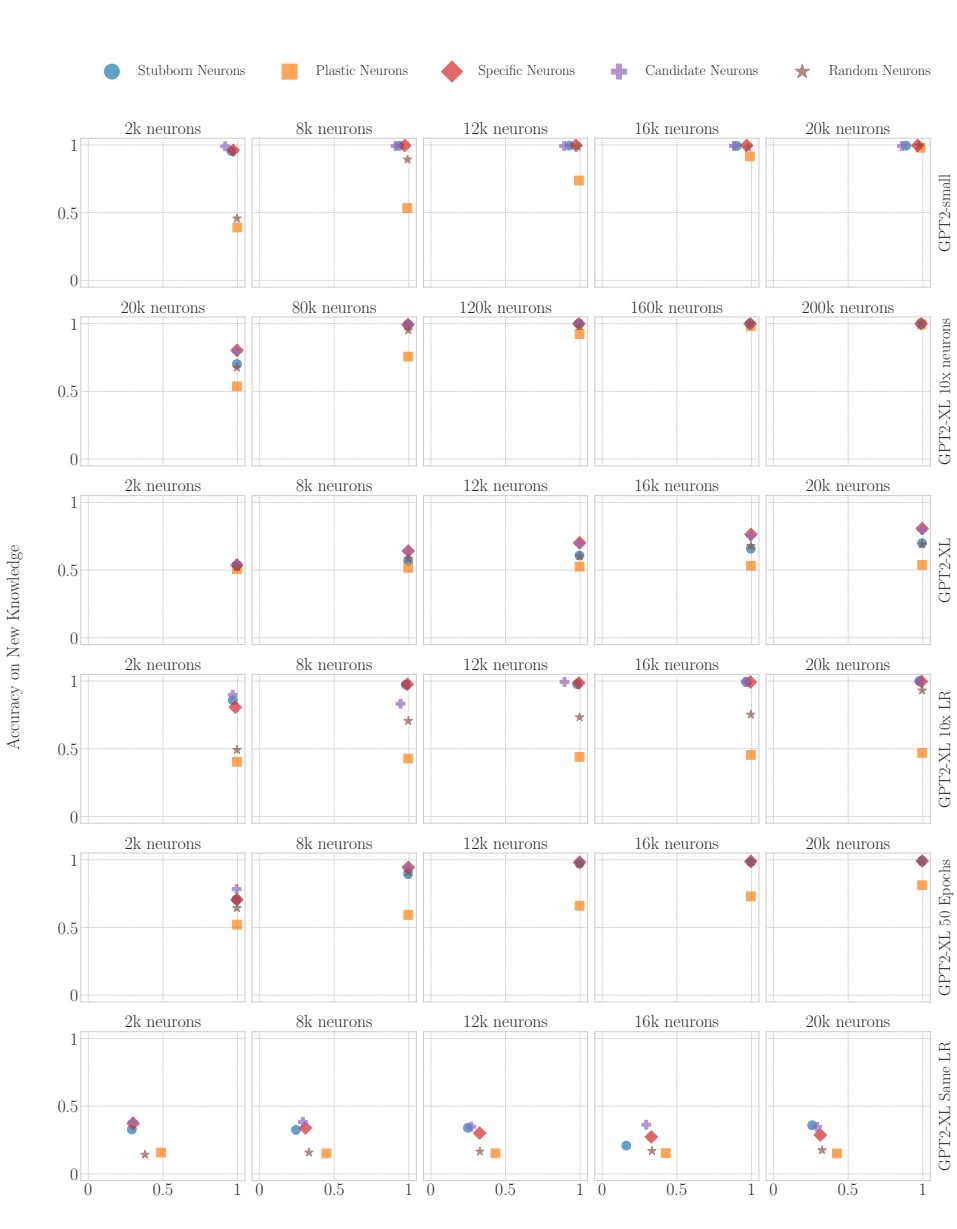

Figure 12: **Non-dissonant update.** Scatter plot of old (x) vs new (y) knowledge during incremental updates with new knowledge for different strategies and scopes (N). gpt2-small (top row) and gpt2-xl (bottom row) with combined variations including 10x neurons, 10x learning rate, 50 epochs, and same learning rate for gpt2-xl.

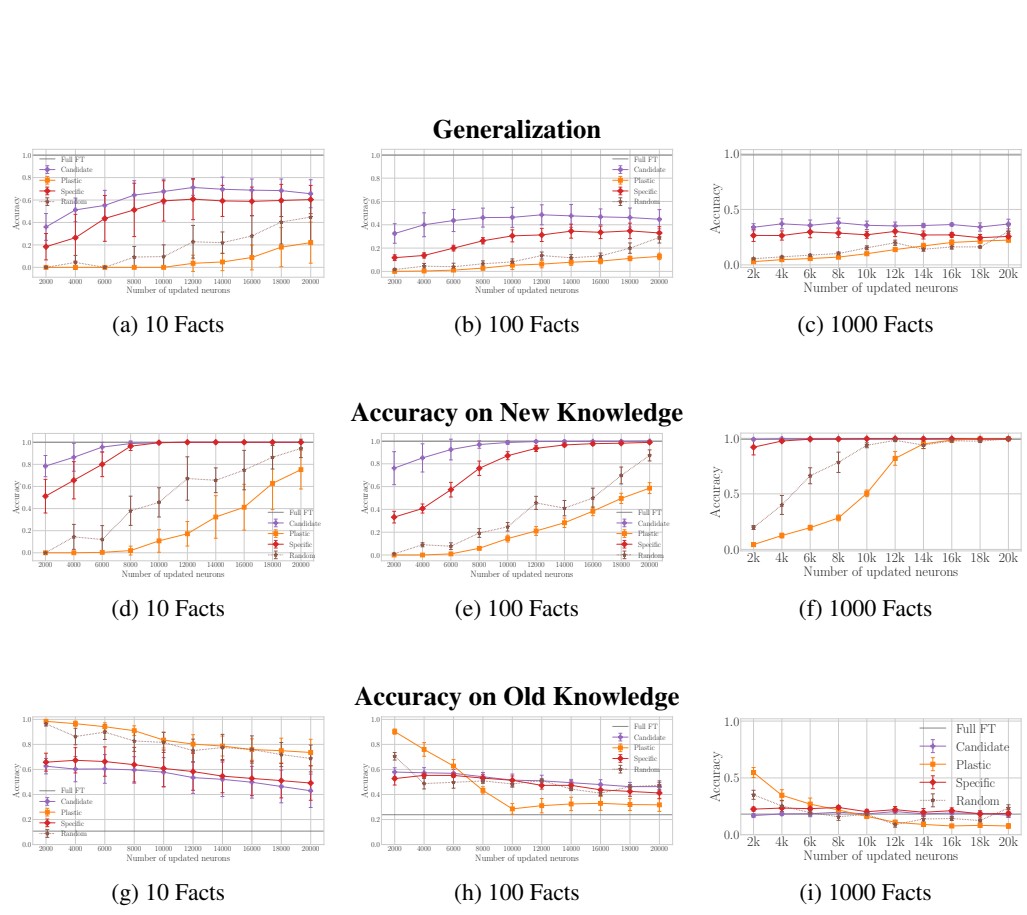

Figure 13: Performance metrics of gpt2-small during dissonant knowledge update, across different numbers of conflicting facts. Each row represents a distinct metric: Accuracy on **Generalization**, Accuracy on **New Knowledge**, and Accuracy on **Old Knowledge**. Within each row, the subplots correspond to the number of conflicting facts introduced (**10 Facts**, **100 Facts**, and **1000 Facts**).

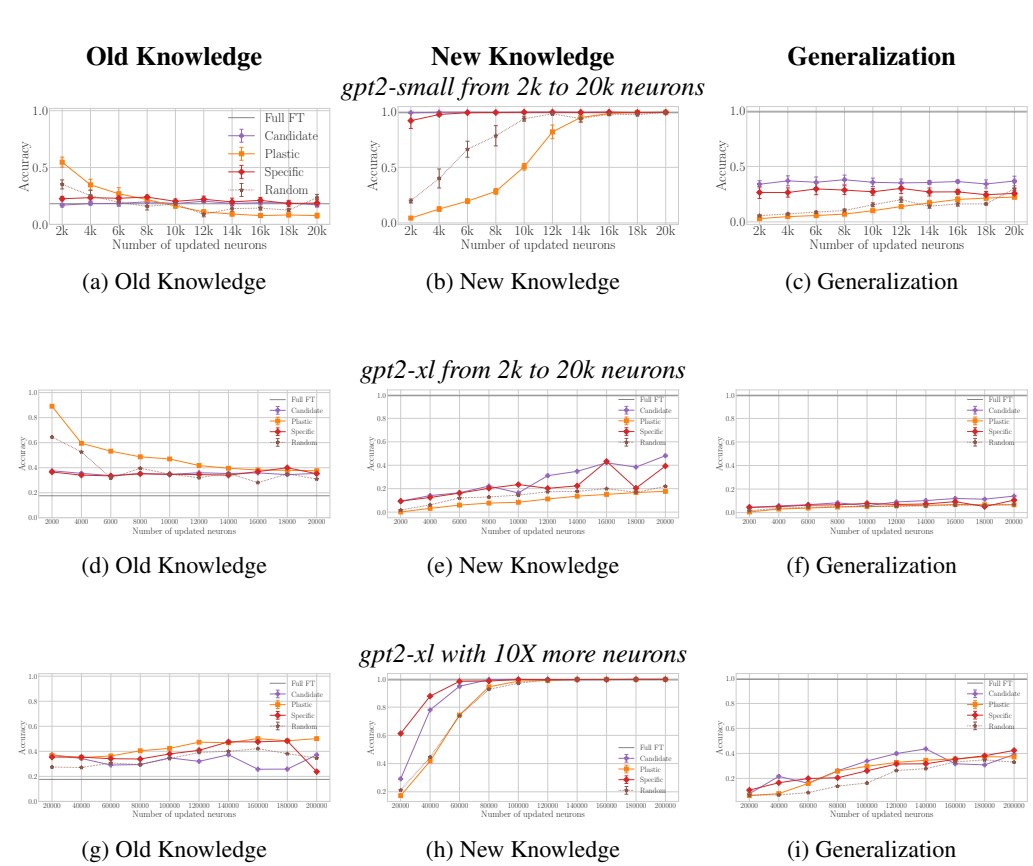

Figure 14: Knowledge Editing Performance of gpt2-xl across different neuron configurations (1000 facts, best learning rate).

