# OpenReview forum: "In Praise of Stubbornness: The Case for Cognitive-Dissonance Aware Continual Update of Knowledge in LLMs"
_ICLR.cc/2025/Conference — Submitted to ICLR 2025_

### Official Review · Reviewer_fE4z · 2024-11-04

**Soundness:** 4
**Presentation:** 2
**Contribution:** 3
**Rating:** 6
**Confidence:** 4

**Summary:**

This paper introduces a novel approach to help large language models (LLMs) update their knowledge without catastrophic forgetting, inspired by human cognitive processes. The authors propose two main components: (1) a mechanism to classify new information as novel, familiar, or dissonant using activation and gradient patterns, and (2) a targeted update strategy that identifies "stubborn" (frequently used) and "plastic" (rarely used) neurons based on historical gradient information. Through experiments on GPT-2 models, they show that their approach can effectively classify information types and that targeted updates help preserve existing knowledge when incorporating non-conflicting information. However, they find that updating with conflicting (dissonant) information remains challenging and destructive to prior knowledge, even with their targeted approach.

**Strengths:**

The paper introduces several ideas that are quite original; they make insightful use of inspiration from human cognitive abilities, such as dissonance and familiarity awareness, and the innovative application of these ideas to LLMs for continual updates. The introduction of dissonance awareness as a mechanism for classifying information into novel, familiar, or conflicting is cool.

 The use of historical activation and gradient information to classify neurons into "plastic" or "stubborn" is interesting. The use of activation and gradient patterns for dissonance detection is also interesting.

Altogether, this study aims to contribute to a critical challenge in LLM deployment (continual knowledge updates).

**Weaknesses:**

Nevertheless, this paper has a number of major shortcomings.

First and foremost was that their paper was at times quite unclear and difficult to read. Procedures for their analysis e.g. how they classified input information, were disconnected and took a substantial amount of time to piece together. The authors should be encouraged to better organize their methods in a more optimal way so that the readers can have a more crystal clear picture of what they did.

1. Assuming I have pieced together their method correctly: their definition of dissonance awareness does not actually come from some unsupervised procedure on the activation / gradient patterns alone. They use the COUNTERFACT dataset which already contains labeled pairs of facts and their contradictions to train their classifier. To what extent is such a classifier works at all, out of distribution, is uninvestigated and is a limitation if further experimentation is not done. I highly encourage the authors to do this because their idea of this classifier is one of the main novelties of their paper.

From what I can gather: the authors did generate "new" facts using GPT-3.5 but this, too, was done by modifying COUNTERFACT facts and is therefore quite in-distribution.

2. From the paper, they developed 4 different strategies for selecting which neurons to update when learning new inputs. The authors claim that their strategies beat state of the art, but it is apparently from the results that the baseline results are already quite saturating even without any careful selection of neurons to update.

Moreover, all 4 strategies struggle in the dissonant information case, the most interesting learning scenario by far, so this scenario remains unsolved even with their 4 strategies.

Is there a non-saturated scenario in which the 4 strategies outperform other methods?

Having said this, I do hope that the authors can find an interesting use-case for their method as I do find their method to be quite potentially interesting (and inspired by human cognition). Or at least, to uncover an interesting empirical model behavior to report to the community when utilizing their method or their classification. As it stands, both can be improved.

**Questions:**

Please address the 2 points in weakness above.

---

> ### Author Response · Authors · 2024-11-21
> **Answer to reviewer fE4z 1/4**
>
> Thank you for your review and motivating comments.
>
> > First and foremost was that their paper was at times quite unclear and difficult to read. Procedures for their analysis e.g. how they classified input information, were disconnected and took a substantial amount of time to piece together. The authors should be encouraged to better organize their methods in a more optimal way so that the readers can have a more crystal clear picture of what they did.
>
> Thank you for highlighting this important clarity issue. You're absolutely right that the connections between components weren't sufficiently clear. To address this, we are adding a comprehensive diagram that clearly illustrates our experimental pipelines:
>
> The diagram [(see link)](https://imgur.com/a/jpQx7H5) exemplifies our two key experimental setups:
>
> 1. Non-dissonant Updates (Left):
> - Start with pretrained model
> - Update with 2000 old facts, collecting *historical* activations & gradients, and hence identifying stubborn (and free neurons)
> - Simulate updates with 1000 new facts and collect *current* activations & gradients to identify candidate neurons,
> - Apply different update strategies (candidate, specific, stubborn, plastic, random)
> - Track performance on both old and new knowledge
>
> 2. Dissonant Updates (Right):
> - Same initial setup with 2000 old facts
> - Update similarly with 1000 new facts
> - Then introduce 1000 updates conflicting the last 1000.
> - Compare strategies' ability to handle conflicts while preserving unrelated knowledge (2000 old facts)
>
> This visualization helps connect how historical gradients inform neuron classification (stubborn vs plastic) and how we leverage this for targeted updates. It also clarifies on which unrelated dataset the catastrophic forgetting happens in the case of dissonant updates (the original old 2000 facts).
>
> We hope that this revised structure better emphasizes our key empirical findings:
>
> 1. LLMs show remarkable robustness to non-dissonant updates (when avoiding stubborn neurons)
> 2. Dissonant updates cause catastrophic forgetting of unrelated knowledge (regardless of method, even if we avoid stubborn neurons)
> 3. Simple features can effectively detect dissonant information before updates occur
>
> These findings suggest we need to fundamentally rethink how we handle conflicting information in LLMs - perhaps moving from direct editing toward maintaining and contextualizing potentially conflicting knowledge, similar to how humans handle updated information (like maintaining both old and new knowledge about Pluto's planetary status).

---

> ### Author Response · Authors · 2024-11-21
> **Answer to reviewer fE4z 2/4**
>
> > Their definition of dissonance awareness does not actually come from some unsupervised procedure on the activation / gradient patterns alone. They use the COUNTERFACT dataset which already contains labeled pairs of facts and their contradictions to train their classifier. To what extent is such a classifier works at all, out of distribution, is uninvestigated and is a limitation if further experimentation is not done. I highly encourage the authors to do this because their idea of this classifier is one of the main novelties of their paper. From what I can gather: the authors did generate "new" facts using GPT-3.5 but this, too, was done by modifying COUNTERFACT facts and is therefore quite in-distribution.
>
> Thank you for highlighting the concerns about our classifier's setup. Let us clarify our methodology with help of the figure we plan to add to the paper [(please see link)](https://imgur.com/a/ztb1imy).
>
> Our classifier training pipeline consists of two main stages:
>
> 1. Training Data Collection:
> We build a balanced dataset from three sources using CounterFact as starting point:
> - Dissonant facts: Directly sourced from CounterFact's counter-facts
> - Known facts: We use two approaches:
>   * Facts already known by pre-trained GPT2
>   * Facts taught to GPT2 through fine-tuning
> - Unknown facts: Generated by GPT3.5 by transforming CounterFact facts while preserving structure
>
> 2. Classifier Training:
> - Extract activation/gradient features from both pre-trained and fine-tuned GPT2
> - Train SVM/RF classifiers using 80/20 split with 5-fold cross-validation
>
> It's important to note that all our samples (known, unknown, and dissonant) share similar structure, as they are all derived from or inspired by CounterFact's format. This actually makes the classification task challenging - the classifier must learn to distinguish between these categories based on the model's internal representations rather than relying on surface-level structural differences.
>
> Our goal was to demonstrate the feasibility of dissonance detection using simple activation and gradient features in a controlled setting. While we acknowledge that future work could explore broader types of knowledge beyond factual statements, our current results (95-99% accuracy) show that LLMs do encode distinguishable signatures for familiar, novel, and dissonant information - even when these share similar surface structure.
>
> This finding suggests the potential value of integrating dissonance detection into LLM training pipelines to identify and properly handle conflicting information before it causes catastrophic forgetting of unrelated knowledge.

---

> ### Author Response · Authors · 2024-11-21
> **Answer to reviewer fE4z 3/4**
>
> > From the paper, they developed 4 different strategies for selecting which neurons to update when learning new inputs. The authors claim that their strategies beat state of the art, but it is apparently from the results that the baseline results are already quite saturating even without any careful selection of neurons to update. Moreover, all 4 strategies struggle in the dissonant information case, the most interesting learning scenario by far, so this scenario remains unsolved even with their 4 strategies.
>
> Thank you for this observation - you've also helped us realize we need to better position our work's contributions. You're indeed right on both counts, and that's exactly what we highlight in our paper's key takeaways: baseline performance is already quite good for *non-dissonant* updates, and none of our strategies (nor existing methods) truly solve the *dissonant* case.
>
> This is actually one of our key empirical findings:
>
> 1) LLMs show remarkable robustness when incorporating non-dissonant information, regardless of strategy (as long as we avoid stubborn neurons). While our targeted strategies show some efficiency advantages (requiring fewer neurons), the high baseline performance reveals an important property of how these models handle non-conflicting updates.
>
> 2) Our more critical finding emerges with dissonant updates: here, ALL approaches - our strategies/baselines, and existing editing methods - fail catastrophically, affecting even completely unrelated knowledge (the original 2000 facts). This suggests a fundamental limitation in how neural networks handle contradictions, one that persists across model sizes.
>
> This leads us to believe we may need to fundamentally rethink how we approach knowledge updating in LLMs. Rather than trying to "edit" or overwrite weights, perhaps we need mechanisms that can maintain and contextualize conflicting information - similar to how humans maintain both old and updated knowledge (like remembering both Pluto's historical and current planetary status).
>
> We apologize that our initial framing about "beating quantitatively" state-of-the-art editing methods obscured these more important findings. Our work's value lies not in proposing superior editing strategies, but in revealing fundamental properties about how LLMs handle knowledge updates and the need for new approaches to handling conflicting information. Our work actually even tentatively suggests to abandon editing all together (we will better reflect this in the implications section of our work).
>
> > Is there a non-saturated scenario in which the 4 strategies outperform other methods?
>
> Your question about non-saturated scenarios was one of those which revealed an important unclarity from our side. Our goal wasn't finding scenarios where our strategies outperform others - it's instead about what we can actually measure and verify in larger models.
>
> Having in mind our experimental pipeline [(link to diagram)](https://imgur.com/a/jpQx7H5), in our experiments, we deliberately track just 2000 initial facts (proxy for old knowledge). With GPT2-XL's larger capacity, our non-dissonant updates *appear* "successful" because they don't interfere with these tracked facts. But this "success" is deceptive - we have no way to measure what other pre-trained knowledge we might be erasing. Unlike with GPT2-small, where we can control and observe the full extent of relevant knowledge, with larger models we're essentially operating blindly regarding their broader knowledge base.
>
> This is precisely why we focused on controlled experiments with smaller "saturated" models - not to achieve better metrics, but to fundamentally understand how LLMs handle knowledge updates. This controlled setting revealed something, we believe, far more important than performance comparisons: even in larger models, dissonant updates remain catastrophically destructive to existing knowledge. This suggests we're observing fundamental properties of how neural networks handle knowledge conflicts, rather than issues that can be solved by better update strategies.
>
> Finally, rather than asking which scenarios allow our strategies to outperform others, we ended up asking ourselves whether our current approach to knowledge updating in LLMs - attempting to overwrite weights in the case of conflicting updates - is fundamentally misaligned with how knowledge should be updated and maintained (incremental integration instead of knowledge erasure).
>
> We believe the misconception to come mostly from some superficial formulations in our abstract and introduction, which we revised accordingly (see our main comment to all reviewers).

---

> ### Author Response · Authors · 2024-11-21
> **Answer to reviewer fE4z 4/4**
>
> > Having said this, I do hope that the authors can find an interesting use-case for their method as I do find their method to be quite potentially interesting (and inspired by human cognition). Or at least, to uncover an interesting empirical model behavior to report to the community when utilizing their method or their classification. As it stands, both can be improved.
>
> Thank you very much for this final encouraging perspective! Your comment actually touches on what we realize is the most profound implication of our work - one that goes beyond specific methods to fundamental insights about LLM behavior.
>
> Our investigation, inspired by human cognition, revealed three critical empirical findings about how LLMs handle knowledge updates:
>
> 1. LLMs demonstrate remarkable robustness when incorporating non-dissonant information: *as long as we avoid highly-trained (stubborn) neurons, they naturally preserve existing knowledge*. This fundamental property is clearest in our controlled experiments with smaller models, where we can track all relevant knowledge. With larger models like GPT2-XL, while our tracked facts are preserved, we have no visibility into potential effects on other pre-trained knowledge.
>
> 2. However, when faced with dissonant (conflicting) information, LLMs exhibit catastrophic forgetting of completely unrelated knowledge - a behavior that persists even in larger "non-saturated" models like GPT2-XL! This suggests we're observing a fundamental limitation in how neural networks handle contradictions.
>
> 3. Most promisingly, we discovered that LLMs encode detectable signatures of novelty and dissonance in their aggregated activations and gradients (and perhaps in the future even simpler features as suggested by another reviewer).
>
> We believe that our findings point to something far more interesting than a new editing method: they suggest we may need to fundamentally rethink how we approach knowledge updating in LLMs. Incremental non-dissonant knowledge can be ingested, but more care needs to be made with dissonant updates. Just as humans don't erase their understanding that "Pluto was once considered a planet" when learning its current classification, perhaps LLMs need mechanisms to maintain and contextualize potentially conflicting information rather than attempting to overwrite it.
>
> Our classifier could be a first step toward this goal - potentially allowing us to detect and properly handle conflicting information during both pre-training and deployment, before it causes catastrophic forgetting. We believe this opens exciting new directions for making LLMs more cognitively plausible in how they handle evolving knowledge.

---

> ### Comment · Reviewer_fE4z · 2024-11-27
>
> I can see the excellent intellectual effort the authors have put in to come up with these alternative and interesting viewpoints in response to the weaknesses I raised with their original framing of the paper. I see that they have rewritten large portions of their paper around these alternative and interesting viewpoints.
>
> Future work could develop experiments around these alternative and interesting viewpoints for a heightened paper.
>
> As it stands, I definitely recommend acceptance of the paper and have raised my score. Good work.

---

> > ### Author Response · Authors · 2024-12-01
> >
> > We thank you very much for your openness to our arguments and new positioning. We have updated the paper (and global answers) accordingly.

---

### Official Review · Reviewer_2NWd · 2024-11-05

**Soundness:** 2
**Presentation:** 3
**Contribution:** 2
**Rating:** 3
**Confidence:** 4

**Summary:**

This paper studies how new information can be learned by different sets of neurons in LMs. The paper divides facts a model might encounter into novel, familiar, and dissonant categories. These categories can be automatically detected via classifiers that take in statistics about model activations/gradients on the new data as well as historical activations/gradients from earlier model training. Once a new fact's category is identified, the same activation/gradient statistics can be used to categorize model neurons into stubborn or plastic categories. In particular, the paper proposes to learn new facts in plastic neurons (neurons with high gradients for the new fact) that are not already in the “stubborn” category (neurons with high gradients historically). Experiments aim to show how limiting optimized model parameters to these neurons (weights corresponding to these neurons?) leads to more effective learning of new information while retaining old known information. Experiments are conducted with GPT2-small and GPT2-xl on CounterFact data augmented with LLM-generated data. Results show that it is good to avoid using stubborn neurons to learn new facts, although randomly selecting neurons performs about as well as any other method in practice.

**Strengths:**

- Important: The core ideas in the paper are quite interesting. Model editing and continual learning may benefit from first classifying information as novel/familiar/counterfactual in order to decide how to best handle incorporating it into the model. Localizing gradient updates into particular weights or subspaces has proved successful historically, and it stands to reason that more interpretability results could help improve the localization further.
- Important: The choice of datasets is reasonable, and the plots communicate the results well. The experiments properly test the main claims in the paper.

**Weaknesses:**

- Very important: The results sections, as well as the abstract/into/discussion, tend to gloss over the fact that the random neuron selection does quite well compared to any other selection method. I can buy that one should be avoiding stubborn neurons — that seems empirically supported. But it seems like the rest of the positive claims in the paper fall through based on this result regarding random selection. In particular, both Fig 3 and Fig 4 show that random neuron selection does very well, and I feel like this undermines the core claims of the paper. My remaining uncertainty here concerns the fact that you can do well with fewer neurons. But then eventually the random neurons selection catches up, and can even perform favorably to specific/candidate neurons in some respects (like maintaining old knowledge, in some subplots). What are we to make of this?
- Important: Why try to do activation/gradient classification to determine what facts are new or old to the model? There is a simple missing strategy here, which is to use model probabilities for p(o|s,r) and p(”s r o”), obtained from the LM. These features should be sufficient to distinguish between novel, dissonant, and familiar statements. Now, one might say that the strategy in this paper is reasonable, and this is simply an alternative. But the issue is that using model probabilities as features is much easier and more scalable than using (historical) activation and gradient statistics. And it undermines the surprise of the main claim, which is that these properties of the facts are detectable from model internals. It can be entirely unsurprising that these fact properties are detectable from model internals when some of the entities in the facts are made-up, like made-up names, languages, or places. These properties would be detectable from surface features of the text alone, and therefore would likely be detectable based on hidden representations as well. Do the models represent things that a simple probe would not? I am not sure (see: https://aclanthology.org/2021.emnlp-main.122/).
- Important: The paper conducts most of its analysis on GPT-2 small and GPT-2XL, which are not very representative models these days for work on knowledge editing.
- Important: There’s another overclaim in the Discussion, about MEMIT. One of the proposed methods gets a better harmonic mean score than MEMIT. These results aren’t really comparable because they are at very different points on the Pareto curve of new vs old knowledge accuracy. It is hard to say this is an apples to apples comparison. Different scales of the metrics would produce different harmonic means favoring one method or the other.
- Of some importance: The paper is unclear about some details. See questions.

**Questions:**

- Where do the labels for novel/familiar/counterfactual data come from? More information on this dataset for classifier training would be helpful.
    - Can you specify whether the novel facts use new subjects, relations, and/or objects?
- I feel that the feature importance section doesn’t add much to the current draft (and it seems like the authors might agree it is not central).
- The abstract doesn’t really need to mention future work.
- I feel that the limitations section in the introduction does not add much to the paper, besides trying to pull back some of the confident claims in the abstract/intro.
- In 2.1 you could specify what training process you’re talking about. Is it pretraining?
- To select plastic neurons, do you look for HG near 0, or very negative? I think this is not clear from the notation.
- L.150 - is this a typo “eventually first”?
- L.258 typo: “famliar”
- L.323: “traiend”
- L.506: “outperfors”
- For reference, here is another early work on model editing and continual learning in LMs: https://arxiv.org/abs/2012.00363

---

> ### Author Response · Authors · 2024-11-21
> **Answer to reviewer 2NWd 1/3**
>
> Thank you for your careful review.
>
> > Very important: The results sections, as well as the abstract/into/discussion, tend to gloss over the fact that the random neuron selection does quite well compared to any other selection method. I can buy that one should be avoiding stubborn neurons — that seems empirically supported. But it seems like the rest of the positive claims in the paper fall through based on this result regarding random selection. In particular, both Fig 3 and Fig 4 show that random neuron selection does very well, and I feel like this undermines the core claims of the paper. My remaining uncertainty here concerns the fact that you can do well with fewer neurons. But then eventually the random neurons selection catches up, and can even perform favorably to specific/candidate neurons in some respects (like maintaining old knowledge, in some subplots). What are we to make of this?
>
> Thank you, also, for this important observation about random neuron selection. You've us helped highlighting a presentation issue in our paper - we should have been clearer that our neuron selection strategies (including random) were experimental tools for investigating LLM behavior, not really proposed solutions.
>
> Your observation about random selection's effectiveness (Fig3 and Fig4) is inline with one of our key empirical findings: LLMs show *remarkable robustness when incorporating non-dissonant information, regardless of targeting strategy, as long as we avoid stubborn neurons* (Note that while random avoids stubborn neurons "by chance", it might be killing other knowledge we are not tracking).
>
> Overall, this wasn't what we initially expected, but it's an important empirical discovery about how these models handle non-conflicting knowledge updates. And that was exactly the summary of Section4 (non-dissonant updates), *as we wrote in its conclusion: "all strategies tend to preserve old knowledge to an extent, including full finetuning."* So while we observed some advantages in efficiency with targeted approaches (fewer neurons needed, as per the lottery ticket phenomenon, as you also rightly noticed), we never claimed these were breakthrough improvements. We hope the new positioning clarifies this aspect.
>
> However, then, a more critical finding of our paper emerges when we look at dissonant (conflicting) updates (Fig5): here, ALL strategies (including random and targeted) lead to catastrophic forgetting of completely unrelated knowledge (the original 2000 facts that form the prior knowledge of the model, not the knowledge we're currently "conflictually" updating). This suggests a fundamental limitation in how neural networks handle contradictions - one which we found to persist even in a larger model like GPT2-xl.
>
> Finally, this finds an interesting parallel in human cognition: consider how humans handle learning that *Pluto is no longer a planet*: we *don't erase* our previous understanding or knowledge, but maintain both pieces of knowledge with temporal context. Our experiments suggest current approaches to LLM knowledge editing, which attempt to overwrite weights, may be fundamentally misaligned with this more sophisticated cognitive approach.
>
> *Our paper is a call to reconsider continual update of knowledge with the lens of conflict detection and conflict resolution, rather than editing or continual learning alone.*
>
> We are sorry that this presentation mistake diverts the readers from the key findings of our work. We updated the abstract and introduction to reflect these core findings and tune down the competitive comparison with the state of the art.

---

> > ### Comment · Reviewer_2NWd · 2024-11-25
> >
> > Thanks for the response!
> >
> > > we should have been clearer that our neuron selection strategies (including random) were experimental tools for investigating LLM behavior, not really proposed solutions.
> >
> > This framing makes sense to me! Then to my understanding, the main finding is that historical and current gradient information can point to neurons that can best integrate a new fact into the model.
> >
> > > a more critical finding of our paper emerges when we look at dissonant (conflicting) updates (Fig5): here, ALL strategies (including random and targeted) lead to catastrophic forgetting of completely unrelated knowledge
> >
> > I have a concern here that I don't think came up previously. Results on larger models with model editing methods like ROME / LoRA show that it is easy to avoid forgetting completely unrelated knowledge. This makes me question if the experimental setting in this paper is representative of typical settings in model editing.
> >
> > > Our paper is a call to reconsider continual update of knowledge with the lens of conflict detection and conflict resolution, rather than editing or continual learning alone.
> >
> > I continue to like the core idea here. I think the best demonstration of this idea would involve improving on the performance of a SOTA editing method, by capitalizing on this observation that some facts are totally new and some are conflicting. I don't think the results in this paper are currently the most compelling demonstration of this idea.

---

> ### Author Response · Authors · 2024-11-21
> **Answer to reviewer 2NWd 2/3**
>
> > Important: Why try to do activation/gradient classification to determine what facts are new or old to the model? There is a simple missing strategy here, which is to use model probabilities for p(o|s,r) and p(”s r o”), obtained from the LM. These features should be sufficient to distinguish between novel, dissonant, and familiar statements. Now, one might say that the strategy in this paper is reasonable, and this is simply an alternative. But the issue is that using model probabilities as features is much easier and more scalable than using (historical) activation and gradient statistics. And it undermines the surprise of the main claim, which is that these properties of the facts are detectable from model internals. It can be entirely unsurprising that these fact properties are detectable from model internals when some of the entities in the facts are made-up, like made-up names, languages, or places. These properties would be detectable from surface features of the text alone, and therefore would likely be detectable based on hidden representations as well. Do the models represent things that a simple probe would not? I am not sure (see: https://aclanthology.org/2021.emnlp-main.122/).
>
> We appreciate this insightful suggestion about using model probabilities as an alternative classification approach!
>
> For novelty/familiarity classification, measuring model perplexity alone is indeed likely sufficient (~low surprise meaning known fact). However, detecting dissonance is likely more complex as it requires understanding not just unfamiliarity but actual contradiction with existing knowledge - making it unclear whether model probabilities alone would reliably distinguish between novel and dissonant inputs, in the more general case.
>
> But as you rightly mentioned, our focus was on *demonstrating the feasibility* of dissonance detection, rather than finding the optimal method. While model probabilities offer a simple alternative that definitely deserves to be tested, we used activations and gradients because they were already integral to our targeted update pipeline (*and hence readily available*), allowing us to study both classification and updates within a unified framework.
>
> We are now, however, as much curious as you about whether using model probabilities alone could be enough. As part of our future work, we plan to build a better dataset (than the one based on CounterFact). We think that more work is to come for building better classifiers for more complex text than the simple facts of Counterfact. We will try our best to compare with model probabilities, but we are not sure we would be able to do so in time for the final rebuttal deadline.
>
> So we fully embrace the suggestion about surface-level features versus deeper representations, and we will defintely add a comment about it in the paper.
>
> > Important: The paper conducts most of its analysis on GPT-2 small and GPT-2XL, which are not very representative models these days for work on knowledge editing.
>
> Our use of GPT-2 models reflects a deliberate experimental choice to have better control on what is going on. While recent editing work successfully targets 7B-8B models, our goal differs - we aimed to understand fundamental mechanisms of knowledge updating.
>
> Please see our diagram [(see link)](https://imgur.com/a/jpQx7H5) to rememorate our experimental pipeline: Our controlled experiments track how models learn 2000 initial facts, then 1000 new facts (non-dissonant updates), followed by conflicting updates to these 1000 facts (dissonant updates).
>
> With GPT2-XL's larger capacity, the probability of interfering with our 2000 tracked facts becomes even smaller - which we confirm experimentally. However, this apparent "success" is deceptive: we can't see what other pre-trained knowledge we might be erasing. More importantly, *even in GPT2-XL, dissonant updates remain catastrophically destructive* - suggesting we're observing fundamental properties of how neural networks handle knowledge conflicts.

---

> > ### Comment · Reviewer_2NWd · 2024-11-25
> >
> > > detecting dissonance is likely more complex as it requires understanding not just unfamiliarity but actual contradiction with existing knowledge - making it unclear whether model probabilities alone would reliably distinguish between novel and dissonant inputs, in the more general case.
> >
> > Right in this case, maybe you can use three features, p(o), p(o|s,r), p("o s r"). I bet conditioning on p(o) would help distinguish between novel and dissonant facts (is the object never-before-seen, or is it familiar but what is surprising is p(o|s,r)?)
> >
> > > While recent editing work successfully targets 7B-8B models, our goal differs - we aimed to understand fundamental mechanisms of knowledge updating.
> >
> > Well, unless you're suggesting that you have compute constraints, it's possible to understand fundamental mechanisms of knowledge updating on 6b-8b models as past work has.
> >
> > > With GPT2-XL...this apparent "success" is deceptive: we can't see what other pre-trained knowledge we might be erasing. More importantly, even in GPT2-XL, dissonant updates remain catastrophically destructive - suggesting we're observing fundamental properties of how neural networks handle knowledge conflicts.
> >
> > I think these points are an accurate characterization of the conclusions one can draw from the experiments, but I worry they are basically limitations of the experiments. I think that other work does a decent job exploring what other pretrained knowledge might be lost from knowledge updates, for instance by looking at local/neighboring facts. Plus, see the above comment on catastrophic forgetting for random facts.

---

> > > ### Author Response · Authors · 2024-11-28
> > > **Additional experiment with output features**
> > >
> > > > Right in this case, maybe you can use three features, p(o), p(o|s,r), p("o s r"). I bet conditioning on p(o) would help distinguish between novel and dissonant facts (is the object never-before-seen, or is it familiar but what is surprising is p(o|s,r)?)
> > >
> > > Thank you, again, for this nice suggestion. We managed to perform a couple of experiments that go in this direction.
> > >
> > > For each fact expressed as a statement involving a subject, relation, and object (e.g., "Danielle Darrieux's mother tongue is French"), we extract features that capture increasing levels of detail about the model's predictions. For this we use as you suggested, both:
> > >   - Conditional probabilities p(o|s,r) at different truncation points*
> > >   - Joint probability p(s,r,o) of the full statement
> > >
> > > The features we extract are as follow:
> > > 1. **Basic Token Probabilities (F1)**:
> > >    - For each of the last N tokens: probability of the actual next token given the first part of the prompt
> > >    - These are simple scalar features capturing model's direct confidence in the correct continuation
> > >    - Dimensionality: 3 + 1 (3 truncation points/tokens, plus full statement p(s,r,o) (likely next token(EoS)) => 4
> > >
> > > 2. **Top-k Predictions Analysis (F2)**:
> > >    - For each position, as above: values and normalized indices of top-k most likely tokens (k=100)
> > >    - Captures both confidence distribution and rankings
> > >    - Computed for the truncated prompts above and the full statement
> > >    - Dim: (N + 1) × 2k (k values and k normalized indices for each position) => 800
> > >
> > > 3. **Distribution Features (F3)**:
> > >    - Full probability distribution binned into histograms
> > >    - Incorporates complete prediction patterns across vocabulary
> > >    - Includes indicator vectors for ground truth tokens
> > >    - Dim: (N + 1) × n_bins (distribution bins for each position =>  400
> > >
> > > 4. **All features (F4)**
> > > - Simple concatenation of all features above.
> > >
> > > *Since the object o can span multiple tokens, we extract features from the last N tokens of each fact (we pick N=three as most answers fit in).
> > > For each token position, we compute both the truncated prompt probability p(o|s,r), as well as the full sentence probability p(s,r,o) (this would correspond likely to EoS, but it does not hurt to keep it and let the classifier decide whether to use it).
> > >
> > > | Strategy (nb_feat) | Pretrained Model | | Finetuned Model | |
> > > |----------|-----------------|-----------------|-----------------|-----------------|
> > > | | Accuracy | F1-Score | Accuracy | F1-Score |
> > > | Feat.1 (4) | 0.852 | 0.856 | 0.850 | 0.855 |
> > > | Feat.2 (800) | 0.602 | 0.588 | 0.600 | 0.581 |
> > > | Feat.3 (400) | 0.540 | 0.452 | 0.543 | 0.464 |
> > > | Concat (1204) | 0.983 | 0.983 | 0.978 | 0.978 |
> > > | A+G (240) | 0.947 | 0.948 | 0.995 | 0.995 |
> > >
> > > As shown in the table, we observe a similar good performance when using the model outputs, compared to Activations and Gradients. Model output achieves even better performance in case of pre-trained models (which is likely the more realistic scenario). This is inline with our earlier observation that activations (what we're using now) are more important than gradients in the case of pre-trained models. The 4 scalar values alone provide already most of the gain (85%). This result is  encouraging for future work, where we plan to (i) build more challenging classification datasets (than the simple facts in CounterFact) and (ii) build standalone classifiers to speed up the training of LLMs, by avoiding training on conflicting data. This is when the comparison of all the methods above will become even more interesting.
> > >
> > > We included these new results to the appendix.

---

> ### Author Response · Authors · 2024-11-21
> **Answer to reviewer 2NWd 3/3**
>
> > Important: There’s another overclaim in the Discussion, about MEMIT. One of the proposed methods gets a better harmonic mean score than MEMIT. These results aren’t really comparable because they are at very different points on the Pareto curve of new vs old knowledge accuracy. It is hard to say this is an apples to apples comparison. Different scales of the metrics would produce different harmonic means favoring one method or the other.
>
> You raise an excellent point about the harmonic mean comparison. We apologize if this came across as claiming superiority over MEMIT - that was not our intent at all. In addition, as we mentioned earlier, we think that our work stands alone, even without the direct comparisons to MEMIT and ROME (initially, we did not even want to include comparisons with Memit/ROME etc to avoid such confusion, but we prefered to put them as *anchor points*).
>
> The message of our paper (and what we've learned along the way) is instead: *what if we were thinking about knowledge updates in LLMs the wrong way? Just as humans don't unlearn their old knowledge when learning something new (imagine again trying to forget that Pluto was once considered a planet. We simply cannot, we still remember such old knowledge), maybe LLMs need mechanisms to maintain and contextualize conflicting information rather than overwriting it* as is done with editing methods today (hence the title's mention to the need of stubbornness and cognitive-dissonance awareness as perhaps fundamental mechanisms and guardrails).
>
> To come back to the detailed comparison, we will highlight that both MEMIT and our approach are operating on different points on the pareto curve, and amend the results section to discuss the trade-offs (We will color Table 3 to better highlight the tradeoffs and remove the "bold" on the strategy with highest harmonic mean).
>
> We're very open to suggestions for better ways to present these results that wouldn't imply direct competition between fundamentally different approaches: we don't want to edit knowledge, we target the problem of continual training of LLMs and what we found to be the two different cases of dissonant and non-dissonant training data.
>
> > - Where do the labels for novel/familiar/counterfactual data come from? More information on this dataset for classifier training would be helpful.
> > - Can you specify whether the novel facts use new subjects, relations, and/or objects?
>
> First, we are adding figures that visually summarize all our experimental settings. For classifier training ([(please see link)](https://imgur.com/a/ztb1imy), we construct a balanced dataset using three sources:
>
> 1. Familiar facts: We explore two approaches:
>    - Finetuned model: Facts explicitly injected into the model through finetuning
>    - Pre-trained model: Facts already known by the model (e.g. 600 facts only identified from pre-training for GPT-2 small)
>
> 2. Dissonant facts: Directly sourced from COUNTERFACT dataset's counterfactuals
>
> 3. Novel facts: Generated using GPT-3.5, maintaining *similar relations* but inventing *new subjects/objects* (examples in Appendix B.1 and below*). We decided to keep similar relations so that the "Novel facts" are *not* completely different from the known facts.
>
> We use 5-fold cross-validation with 80/20 splits to evaluate classifier performance.
>
> * example of generated novel facts:
> ```
> - Original fact: “Toko Yasuda produces the most amazing music on the guitar.”
> - Inspired Novel fact: “Zara Zorin produces the most amazing music on the theremin.”
> ```
>
> > - I feel that the feature importance section doesn’t add much to the current draft (and it seems like the authors might agree it is not central).
>
> We think that we failed to make the distinction between the finetuned model and pre-trained model cases clear, which leads to a perceived marginal impact of the feature importance (hopefully clarified now with the new figures).
>
> The feature importance analysis reveals an interesting pattern:
>
> - In the *finetuned case*, where models are trained until convergence on specific facts, gradient features dominate the classification.
>
> - In contrast, for pre-trained knowledge which likely involved less intensive training on any specific fact, activation patterns become more important. This suggests the intensity of training (overfitting vs lighter exposure) affects how distinguishable knowledge becomes in the model, rather than indicating fundamentally different types of knowledge representation.

---

> > ### Comment · Reviewer_2NWd · 2024-11-25
> >
> > > The feature importance analysis reveals an interesting pattern:
> >
> > I think this result is interesting and adds to the paper!
> >
> > ...
> >
> > Based on the rebuttal so far, I would plan on maintaining my score of 3. I think the core idea of the paper is interesting but the experimental results don't demonstrate the idea in a compelling way, largely due to using small models and editing methods (neuron finetuning) that seem far behind the SOTA (low-rank weight finetuning).

---

> > > ### Author Response · Authors · 2024-11-26
> > >
> > > Thank you for the positive tone!
> > >
> > > > I continue to like the core idea here. I think the best demonstration of this idea would involve improving on the performance of a SOTA editing method, by capitalizing on this observation that some facts are totally new and some are conflicting.
> > >
> > > We start with this comment, as we realize the need to clarify a crucial point: editing methods, by their very nature, only handle conflicting updates - they're designed specifically to replace existing knowledge with contradictory information. There is no such thing as "non-conflicting editing" because editing inherently means changing what the model believes. *That's why we think that what we're proposing is fundamentally different*: looking at how models handle both types of updates - conflicting (editing today) and non-conflicting (not present in the literature) - in a continual learning context.
> > >
> > > Next, let's look again at our experimental protocol as it would be useful for the remainder: https://imgur.com/a/jpQx7H5
> > >
> > > > This framing makes sense to me! Then to my understanding, the main finding is that historical and current gradient information can point to neurons that can best integrate a new fact into the model.
> > >
> > > So yes! We believe in addition that two other key findings (perhaps more important) emerged from our investigation:
> > > 1. For non-dissonant updates, we found that preserving stubborn neurons prevents forgetting of old knowledge - supporting the concept of differential plasticity
> > > 2. What really surprised us: even when we preserve stubborn neurons, dissonant updates ("editing") still lead to loss of completely *unrelated* knowledge (see definition of unrelated knowledge below).
> > >
> > > > Results on larger models with model editing methods like ROME / LoRA show that it is easy to avoid forgetting completely unrelated knowledge.
> > >
> > > Thank you for bringing up this really important point! It helps us realize we should have been clearer about a fundamental difference in how we evaluate knowledge preservation.
> > >
> > > You see, **the editing literature, by design, focuses only on dissonant updates (changing facts to counterfacts)**  and looks at how well the model preserves **neighboring** or related facts. And you're absolutely right - methods like ROME/MEMIT are reported to excel at this.
> > >
> > > But our work took us on a different path. We wanted to understand the bigger picture of how LLMs handle knowledge updates more generally, leading us to:
> > > 1. Examine both non-dissonant (new) and dissonant (editing) updates in a continual learning context
> > > 2. Look at the impact on completely **unrelated knowledge**, not just neighboring facts
> > >
> > > And that's how we got surprised: while ROME/MEMIT preserved neighboring facts as discussed in the literature, we found that they significantly impacted completely unrelated knowledge (our 2000 facts which we use, at each new fold, as proxy for old knowledge). This discovery got us really thinking - what if we're approaching knowledge updates in LLMs the wrong way? Looking at how humans handle knowledge updates (like the Pluto planet example above), we don't actually "edit out" old knowledge - we maintain both old and new knowledge with temporal context. What if the solution is to abandon "editing" all together? Just as like humans do?
> > >
> > > To make this distinction clearer in our paper, we'll replace "old knowledge" with "unrelated knowledge". We'll include baseline results on neighboring prompts for comparison.
> > >
> > > > the experimental results don't demonstrate the idea in a compelling way, largely due to using small models and editing methods (neuron finetuning) that seem far behind the SOTA
> > >
> > > We realize now that by comparing our "idea" (strategies) to SOTA editing methods, we might have unfortunately diluted our real contributions. We weren't trying to build a better editing method (as said earlier, we are starting to believe that editing could be a bad idea...). Instead, we wanted to understand how LLMs handle both dissonant ("editing") and non-dissonant (what the literature does not look at yet) updates. We believe that what we found about the impact on unrelated knowledge has important implications for how we think about knowledge updating in LLMs. And this finding was surprisingly visible even when we scaled to GPT2-xl.
> > >
> > > Hence, regarding model scaling: our findings suggest this is fundamentally about the ratio between model capacity and the amount of tracked knowledge, rather than absolute model size. By appropriately scaling the number of facts and counterfacts, we expect similar behavioral patterns would emerge even in larger models - it's about reaching the critical density where knowledge conflicts become observable. This suggests a natural direction for future work: validating these mechanisms at scale while maintaining the careful experimental control that made our current findings possible.
> > >
> > > If our clarifications resonate with you, we'd welcome your advice on how to better convey the findings of our paper.

---

> > > > ### Comment · Reviewer_2NWd · 2024-11-26
> > > >
> > > > Changing the emphasis from model editing to continual learning is fine. I don't think it changes too much about the interpretation of the results in the paper. And I wouldn't say model editing has only looked at counterfactual knowledge. CounterFact may focus on this case, but pretrained LLMs do not even know all the facts in CounterFact, so many of the datapoints (whether factual or counterfactual) are new to them. This is common across datasets used in model editing.
> > > >
> > > > > And that's how we got surprised: while ROME/MEMIT preserved neighboring facts as discussed in the literature, we found that they significantly impacted completely unrelated knowledge (our 2000 facts which we use, at each new fold, as proxy for old knowledge). This discovery got us really thinking...
> > > >
> > > > To be honest, I am basically pretty skeptical of this result. I've worked with model editing methods for a couple years now, and I have always observed, across datasets and models, that random data is easier to retain than neighboring data. I would be very surprised if neighboring data were retrained but random data was not, on average, and I worry that this result may be an artifact of the specific experiment setup in the paper.

---

> > > > > ### Author Response · Authors · 2024-11-28
> > > > >
> > > > > We would like to start with two clarifications:
> > > > >
> > > > > First, when we discuss editing, we of course mean the editing of existing factual associations - the core focus of seminal works like the 2020 one you shared with us, ROME, MEMIT, SERAC, GRACE and much of the popular editing literature. This task is inherently conflictual, fundamentally different from incremental insertion of new knowledge.
> > > > >
> > > > > Second, we did not test yet ourselves the impact on *neighboring knowledge* in our experiment, we just clarified to you that our old knowledge is different from the old "related knowledge" used in editing. This *unrelated knowledge*, which is a novel perspective we propose, shows consistent results observed across different sets of old facts (5 folds of different 2000 facts) and two models (gpt2-small and XL), with each experiment repeated 5 times: on the same model and condition, non-dissonant updates preserve such unrelated knowledge, while dissonant updates destroy it. We find this insightful.
> > > > >
> > > > > That being said, we plan to examine effects on neighboring knowledge in coming days - and it's likely that performance on neighboring knowledge is also degraded in our setting (smaller models, continuous "edits"), similarly to what we observe with unrelated knowledge. Moreover, another key difference between our setting, and what e.g. ROME and many other SoTA editing do, is that we perform sequential updates of all the new facts (while e.g. ROME is usually tested one fact at a time, always restarting fresh from a "virgin" model, which is not a realistic scenario for our continual learning setting; MEMIT on the other hand targetted multiple "edits"). And the literature has already shown how many editing models perform badly when moving to sequential edits, which is the case of our continual learning setting.
> > > > >
> > > > > Finally, we remind that our findings are supported by robust empirical evidence, including multiple experimental runs across different models and fact sets, with full reproducibility through our shared anonymous codebase. **We reiterate that our work and findings stand alone, as a useful empirical investigation, even in the absence of any mention or comparison with existing editing literature. Unlike editing literature which is still on the "research phase", in practice today, LLMs are pre-trained and post-trained, and during these phases, they face conflicting knowledge which at best, slows down their training. Our empirical findings together with appropriate classifiers can remediate this.**
> > > > >
> > > > > Overall, given the very interesting exchanges we had, we strongly believe the community interested in new views on LLMs would be happy to engage with us in ICLR, and would benefit from our work.

---

### Official Review · Reviewer_c7Q3 · 2024-11-05

**Soundness:** 3
**Presentation:** 4
**Contribution:** 3
**Rating:** 8
**Confidence:** 3

**Summary:**

This paper investigates categorizing neurons within large networks based on how often they are used, i.e. how much information they process. Based on this identification the authors propose to target learning novel information, e.g. in a continual setting, to specific sets of these neurons.

**Strengths:**

The paper is refreshingly well written and structured. it is concise and on the point. The idea is very intuitive, however I am not an expert of this field to be able to assess its novelity or relation to previous work.

**Weaknesses:**

I consider it helpful to have a more explicit section dedicated towards relation to previous works. Currently, only the paragraph at ll 78 seems to hint at related works. I think expanding this more (if this is applicable) would help better assess the proposed approach to others and in this way better evaluate its significance. Unfortunately, I can not make real suggestions on what to add though as I am not an expert on this topic.

**Questions:**

In ll 494 the authors write "Language models can classify new information as novel, familiar, or dissonant ...". I am a little confused by this statement. I understood that rather a seperate classifier learns the classification. So it is not the LLMs that can classify. Additionally, (correct me if I'm wrong) the classifier is trained via supervision right? Then this claim/sentence seems a little unspecific.

I am a little concerned about the very different results with the xl model. Do the authors have an intuition on why they are observing these differences? It would be good to add this intuition into the paper more explicitly.

---

> ### Author Response · Authors · 2024-11-21
> **Answer to Reviewer c7Q3**
>
> Thank you for your positive evaluation and thoughtful feedback.
>
> > I consider it helpful to have a more explicit section dedicated towards relation to previous works. Currently, only the paragraph at ll 78 seems to hint at related works. I think expanding this more (if this is applicable) would help better assess the proposed approach to others.
>
> We have reviewed the most recent literature and we are confident in our work's unique positioning at the intersection of editing and continual learning approaches, as synthesized in the taxonomy of Table 1. While existing work either focuses on editing individual facts (like ROME and MEMIT) or adapting to new tasks, we follow a different path by studying how LLMs should continuously update their (dissonant and non-dissonant) knowledge through human-inspired mechanisms.
>
> Our empirical findings reveal critical insights that neither editing nor continual learning literature has addressed:
>
> 1. Dissonant updates cause catastrophic forgetting of unrelated knowledge.
>
> 2. Non-dissonant updates work well when protecting (avoiding) stubborn neurons.
>
> 3. Simple cognitive-inspired classification can effectively detect conflicts.
>
> These findings suggest we may need to fundamentally rethink how LLMs should handle knowledge updates. Like humans who don't erase their understanding that "Pluto was a planet" when learning it's no longer classified as one, perhaps LLMs need mechanisms to maintain and contextualize conflicting information rather than overwriting it.
>
> Said differently, humans do not do editing, instead they incrementally integrate conflicting information and contextualize it [1]. May be we should do the same for LLMs in order to preserve their old knowledge.
>
> While the introduction outlines the above contributions, *we will expand Table 1 with recent literature and add a comprehensive related work section in the appendix to further clarify these distinctions*.
>
> > In ll 494, the authors write "Language models can classify new information as novel, familiar, or dissonant ...". I am a little confused by this statement. I understood that rather a separate classifier learns the classification. So it is not the LLMs that can classify. Additionally, (correct me if I'm wrong) the classifier is trained via supervision right? Then this claim/sentence seems a little unspecific.
>
> Your understanding is correct. Sorry for the confusion and thank you for catching this ambiguity. The sentence should read: "By analyzing LLM activations and gradients with simple supervised classifiers, we can classify new information as novel, familiar, or dissonant with roughly 95% (resp. 99%) accuracy for pre-trained (resp. finetuned) models."
>
> > I am a little concerned about the very different results with the xl model. Do the authors have an intuition on why they are observing these differences? It would be good to add this intuition into the paper more explicitly.
>
> We apologize for the lack of clarity: we are adding a diagram [(see link)](https://imgur.com/a/jpQx7H5) illustrating our experimental setup, which should help clarifying our findings with GPT2-XL (which we will also enhance in the paper explicitly as you rightly suggest).
>
> Our experiments use 2000 facts as a proxy to measure "old knowledge". These 2000 facts represent:
> - A significant portion of GPT2-small's parameter space
> - A much smaller fraction of GPT2-xl's capacity
>
> This difference in relative capacity explains some of our key findings:
>
> For non-dissonant updates:
> - With only 2000 tracked facts in GPT2-xl's larger parameter space, knowledge interference is naturally less likely
> - This explains why protecting stubborn neurons shows less benefit - there's simply more room to store new knowledge without conflicts with the 2000 old facts
> - However, we discovered that effective learning in GPT2-xl (using our targetted strategies which use fewer neurons) requires either:
>   * Longer training time (more epochs), or
>   * Higher learning rates compared to the finetuning baseline and compared to gpt2-small. This scaling behavior is itself an interesting empirical finding (that deserves attention in future work).
>
> Importantly here, while performance metrics on our 2000 tracked facts look good, *we have no visibility into what other pre-trained knowledge inside GPT2-XL might be erased* - the model could be catastrophically forgetting important information we cannot measure.
>
> For dissonant updates:
> - *Despite the larger* capacity, *conflicting updates remain catastrophically destructive* to existing knowledge, even when avoiding stubborn neurons.
> - This suggests the destructive nature of dissonant updates is a fundamental property of how these models handle contradictions, not just a capacity issue
>
> We will revise the paper to make these intuitions explicit and better organize the GPT2-xl results. We have added the experimental setup diagram and will expand the discussion of these scaling effects.

---

### Official Review · Reviewer_h23s · 2024-11-07

**Soundness:** 3
**Presentation:** 3
**Contribution:** 3
**Rating:** 6
**Confidence:** 3

**Summary:**

The authors present a detailed framework for continual learning in LLMs through targeted neuron updates. Their framework incorporates an external dissonance-aware classifier that uses neuron activations and gradients based features to categorize input text as familiar, novel, or contradictory. They introduce various strategies — plastic neurons, stubborn neurons, candidate neurons, and specific neurons — to select neurons for targeted LLM updates.

Experimental results on the COUNTERFACT dataset using GPT-2-small and GPT-2-xl demonstrate that the proposed framework can accurately classify input data as familiar, novel, or contradictory using simple classifiers like SVM and Random Forest. Detailed experiments on targeted updates show that the framework effectively updates LLMs for both conflicting and non-conflicting data, outperforming baseline methods.

An anonymized codebase is provided for reproducibility.

**Strengths:**

- The method, inspired by high-level human brain mechanisms for knowledge updates, is novel and well-suited for LLMs.
- The paper is well-written, with a clear motivation and goal, along with thorough explanations of the framework's technical details.
- Experimental results on GPT-2-small and GPT-2-xl provide valuable insights, highlighting the advantages of considering plastic and stubborn neurons for continual LLM updates.

**Weaknesses:**

- The experimental scope is limited to two relatively small LLMs — GPT-2-small and GPT-2-xl. As mentioned in the paper, the framework performs better on the smaller GPT-2-small model compared to GPT-2-xl. Results on larger models, such as 7B or 8B parameter LLMs, would provide a more comprehensive evaluation of the framework's effectiveness.
- The conflicting information update experiments (Section 3.4) include only a few baseline comparisons. Expanding the evaluation with additional baselines, such as PMET or GRACE, would strengthen the analysis.

**Questions:**

See the Weakness part

---

> ### Author Response · Authors · 2024-11-21
> **Answer to Reviewer h23s**
>
> First, thank you for the encouraging comments on the strengths of our work. Below are our detailed answers to the weaknesses.
>
> > The experimental scope is limited to two relatively small LLMs — GPT-2-small and GPT-2-xl. As mentioned in the paper, the framework performs better on the smaller GPT-2-small model compared to GPT-2-xl. Results on larger models, such as 7B or 8B parameter LLMs, would provide a more comprehensive evaluation of the framework's effectiveness.
>
> Our use of smaller models was intentional for experimental control. With GPT-2-small, we can observe more easily how updates affect old knowledge, since the 2000 old facts we use as a proxy to measure old knowledge represent a more considerable part of the model's parameter space. (please see https://imgur.com/a/jpQx7H5 to recall more easily our experimental pipeline and what we use as proxy for old knowledge).
>
> With GPT-2-xl, while performance metrics on our 2000 tracked facts look better, *we have no visibility* into what other pre-trained knowledge (other than the 2000 facts we're tracking) might be erased - the model could be catastrophically forgetting important information we cannot measure. This blind spot makes larger models less suitable for studying these fundamental phenomena (with a dataset as small as CounterFace and its 17k facts only, of course).
>
> Importantly, even in GPT-2-xl (when non-dissonant updates preserve almost all old knowledge, regardless of the strategy), *dissonant updates remain catastrophically destructive to our selected pre-existing knowledge*. This is valid, even when we completely avoid the stubborn neurons that encode prior knowledge, which is, we believe, a strong empirical finding.
>
> We reformulated our position to better clarify our intentions and results.
>
> > The conflicting information update experiments (Section 3.4) include only a few baseline comparisons. Expanding the evaluation with additional baselines, such as PMET or GRACE, would strengthen the analysis.
>
> First, we will do our best to add at least one more baseline to our comparison (GRACE since its GPT2-compatible with the Easyedit framework).
>
> That being said, we believe that our experimental results point to a more fundamental issue: none of the methods, neither our strategies nor editing, solve the problem of conflicting/dissonant updates. The way our experiments were designed allowed us for better measurements of the impact of editing on prior knowledge, highlighting these limitations in two of the most popular editing methods (ROME and MEMIT). Non-dissonant updates, however, pose no problem, as long as we avoid the stubborn neurons.
>
> This suggests that we may be approaching knowledge updating in LLMs from the wrong angle. Consider how humans handle conflicting information - we don't unlearn our outdated knowledge (which would be catastrophic), but rather maintain it while learning that it should no longer be used. For example, when learning that Pluto is no longer classified as a planet, we don't erase our previous understanding - we maintain both pieces of knowledge along with their temporal context.
>
> Current LLM editing methods, by directly modifying weights, more closely resemble forced forgetting than human-like knowledge updating. Our empirical findings, particularly the catastrophic nature of dissonant updates, suggest we might need fundamentally different approaches that can maintain and contextualize conflicting information rather than overwriting it. This may require rethinking not just how we edit models, but how we represent and update knowledge in LLMs altogether.
>
> We believe these insights, while demonstrated on smaller models (GPT-2-small and XL), reveal important principles about LLM knowledge representation and updating that merit further investigation at larger scales (but would need novel datasets and perhaps methodologies).
>
> So unlike editing methods which focus on replacing individual facts, our work addresses the broader challenge of continuous knowledge integration in LLMs, showing that detecting and handling dissonant information is crucial for successful updates.

---

### Author Response · Authors · 2024-11-21
**Summary of answers and implemented revisions 2/2**

**4. Choice of Model Size and Experimental Control**

Some reviewers questioned our use of smaller models (GPT2-small and XL). *This was a deliberate choice for experimental control*. With GPT2-small, our 2000 tracked facts represent a significant portion of the model's knowledge, allowing us to *better observe update effects*.

While GPT-2-XL shows minimal interference with our tracked facts during *non-dissonant* updates, this apparent robustness is deceptive - we cannot see what other pre-trained knowledge might be erased. *More importantly, with GPT-2-XL, dissonant updates remain catastrophically destructive* (50-80% loss of prior knowledge), *confirming our small model results*.

📝 🧪 We clarified the above, hiding unnecessary complexity*** in scaling behaviors, and modified Fig.6 to compare GPT2-small against a GPT2-XL model trained 10 times longer to make sure it integrates new knowledge.

*** We indeed discovered that while it is relatively easy to pack 1000 facts in 2000 neurons on GPT2-small, it takes (roughly) 10X longer or 10X more neurons to do the same thing on GPT2-XL. This scaling behavior, while interesting, diverts from our main investigation.

**5. Dissonance Classifier Implementation and Dataset**

On classifier robustness (Reviewers 3 and 4), we acknowledge the current limitations of our CounterFact-derived dataset. However, our experimental design was *intentionally restrictive*: by maintaining similar structure across all samples (known, unknown, and dissonant facts), we deliberately made the classification task more challenging. If we had used differently structured facts for each category, the classifier could have relied on these structural differences alone. Instead, by keeping the structure consistent, we forced the classifier to learn more meaningful distinctions based on how the model processes these different types of information. Our strong results (95-99% accuracy) in this controlled setting suggest that LLMs do encode distinguishable signatures for familiar, novel, and dissonant information, even when these share similar surface form.

📝 Our new figure (https://imgur.com/a/ztb1imy) in Section 3 clarifies our pipeline.

🧪 We also appreciate Reviewer 3's suggestion about trying model probabilities as features. Our choice of activations/gradients was primarily motivated by their availability in our update pipeline and our focus on feasibility. We have now implemented and incorporated such an alternative classifier.

*Given the catastrophic effects of dissonant information we discovered, we believe that developing robust dissonance detection for large-scale training data should become a field priority.*

**6. Implications of our findings and summary of manuscript Revisions**

Our findings suggest we may need to fundamentally rethink how we approach knowledge updating in LLMs. Consider how humans handle evolving knowledge: when we learned that Pluto was no longer classified as a planet, we did not erase our previous understanding but maintained both pieces of knowledge along with their temporal context [1]. Current LLM editing approaches, which attempt to overwrite weights, fundamentally differ from the human cognitive mechanism of updating without erasing.

Our work at the intersection of editing and continual learning suggests *three actionable directions that can transform how we train LLMs today*:

- Pre-training and post-training could benefit from classifiers to detect dissonant data before harmful ingestion.

- New approaches could maintain and contextualize conflicting information rather than overwriting it, as editing methods do today.

- Differentiated plasticity (e.g. varying learning rates between stubborn/plastic neurons) is promising for *non-dissonant* updates.

📝 Sumary of revisions:
- Added experimental pipeline diagrams (section3)
- Wrote an extended related work section in appendix A, augmenting Table 1 with most recent literature.
- Clarified feature extraction pipeline.
- Amended the abstract and introduction to better reflect our contributions.
- Adjusted result presentations to avoid implying competition with editing methods (stressing the nuanced trade-offs between various methods, instead of the single harmonic mean we used earlier).
- Simplifying/clarifying GPT2-XL experiments.

🧪 Additionally implemented reviewer-suggested LLM-output-based classifier.

We are happy that these revisions better communicate our work's true contribution: revealing fundamental properties about how LLMs handle knowledge updates and motivating the adoption of more cognitively-inspired approaches to handling conflicting information.

Sincerely,

The authors

[1] Ross, Lee, Mark R. Lepper, and Michael Hubbard. *"Perseverance in self-perception and social perception: biased attributional processes in the debriefing paradigm."* Journal of personality and social psychology 32.5 (1975): 880.

---

### Author Response · Authors · 2024-11-21
**Summary of answers and implemented revisions 1/2**

Dear reviewers,

We thank you all for your time and constructive feedback. You have helped us significantly improving our paper, and for this, we are grateful. Below is a summary of our rebuttal and revision.

**1. Clarification of Goals and Positioning**

We acknowledge that our initial framing (some claims in abstract and introduction) may have overemphasized comparisons with state-of-the-art editing methods, unfortunately obscuring our paper's most important contributions. We have revised the manuscript to reflect our primary goal and contribution: *a cognitive-inspired empirical investigation of how LLMs handle knowledge updates, revealing novel and fundamental properties, with practical and actionable implications*.

At the core of our investigation lies a fundamental question: how does the placement of new knowledge in an LLM's parameter space impact its ability to learn and retain information? Humans naturally detect and carefully handle contradictory information while easily integrating complementary knowledge. Inspired by this cognitive feature and the brain's adaptive plasticity, we systematically study how LLMs handle these two distinct scenarios - non-dissonant updates (complementary knowledge) and dissonant updates (contradictory information) - by targeting different neural populations, from frequently activated ('stubborn') to rarely used ('plastic') neurons.

**2. Clarification of experimental setting**

📝 Key to our empirical findings is our experimental protocol which was, as rightly commented by a reviewer, unclear. For this, we added two comprehensive diagrams to the main paper (Section 3) to clarify it:

- One illustrating our controlled knowledge update experiments: https://imgur.com/a/jpQx7H5
- One detailing our dissonance detection pipeline (see point 5. below):  https://imgur.com/a/ztb1imy

The left side of the first diagram illustrates our non-dissonant update pipeline:
- Start with pretrained model
- Update with 2000 old facts*, collecting *historical* activations & gradients
- Use them to identify stubborn (heavily used) and plastic (unused) neurons
- Simulate the update of 1000 new facts, collecting *current* activations & gradients
- Use them to identify *candidate* (preferred by gradient backpropagation) and *specific* neurons (candidate avoiding stubborn)
- Apply the different update strategies** to integrate the new facts
- Measure new knowledge learning and old knowledge preservation

The right side shows our dissonant update pipeline:
- Same initial setup with 2000 old facts*
- Update with 1000 new facts
- Introduce 1000 updates conflicting with the last 1000 new facts
- Measure impact on unrelated old knowledge (original 2000 facts)

* The first 2000 facts serve as a proxy to measure the performance on old unrelated knowledge. For robustness, we create 5 different folds with different fact combinations and repeat each experiment 5 times.
** Our experimental toolbox includes six different neuron targeting strategies: stubborn, plastic, candidate, specific, full finetuning, and random placement. *Rather than proposing these as solutions, we use them as experimental probes to understand how knowledge placement affects both learning and retention*.

**3. Key Empirical Findings**

Our systematic investigation revealed several surprising properties of how LLMs handle knowledge updates:

1. *Inherent Robustness to Non-dissonant Updates*: LLMs tend to preserve existing knowledge when incorporating *non-conflicting information*, *especially when avoiding stubborn (heavily-used) neurons*. This explains why random selection performs well, as noted by Reviewer 3 - it's not about the selection strategy but about avoiding stubborn neurons. For example, the random strategy avoids "by chance" the stubborn neurons (which correspond in our experiments only to our old 2000 facts). However, such a strategy may be killing other knowledge that we are not tracking in our experiments. (Anticipating next section, the larger the model, the lower the likelihood to hit those stubborn neurons that we are tracking).

2. *Catastrophic Nature of Dissonant Updates*: *All* the placement strategies we tested fail catastrophically when handling conflicting or dissonant information, affecting even completely unrelated knowledge (up to 80% forgetting). This persists across model sizes, even if we preserve stubborn neurons, suggesting a fundamental limitation in how neural networks handle contradictions, implying that current training approaches may be vulnerable to catastrophic forgetting when training data contains conflicting information.

3. *Feasibility of Dissonance Detection*:  Given the destructive nature of dissonant updates, we show that they can be detected: simple activation and gradient features, readily available in our pipeline, can effectively classify information as novel, familiar, or dissonant (95-99% accuracy).

📝 We rewrote the abstract and introduction, clarifying our positioning.

---

### Meta-Review · Area_Chair_waum · 2024-12-28

**Metareview:**

The reviewers noted that the paper is well-written and the method was interesting, but experimental scope is limited and would benefit form further work in a future submission.

**Additional Comments On Reviewer Discussion:**

NA

---

### Decision · Program_Chairs · 2025-01-22

Reject